# Phytolith profile of *Acrachne racemosa* (B. Heyne ex Roem. & Schult.) Ohwi (Cynodonteae, Chloridoideae, Poaceae)

Priya Badgal[1]*, Poonam Chowdhary[1], Mudassir Ahmad Bhat[1,2]*, Amarjit Singh Soodan[1]

1 Plant Systematics and Biodiversity Laboratory, Department of Botanical and Environmental Sciences, Guru Nanak Dev University, Amritsar, Punjab, India, 2 Department of Botany, Central University of Jammu, Jammu and Kashmir, India

* priyabadgal123@gmail.com (PB); mabhat90@gmail.com (MAB)

**Data Availability Statement:** All relevant data are within the manuscript and its Supporting information files.

## Abstract

*Acrachne racemosa* (B. Heyne. ex Roem. & Schult.) Ohwi of the subfamily Chloridoideae of the family Poaceae is an economically important grass species. Grasses are characterized by deposits of silica in the cells or tissues in the form of phytoliths which protect them from various types of biotic and abiotic stresses. Owing to variable shape and specificity of morphotypes, phytolith helps in taxonomical studies, reconstruction of paleoenvironments and prediction of climate changes. The present study focussed on developing a phytolith profile of the selected species. For isolation of phytolith, Dry Ashing Method was employed, and by epidermal peeling, *in-situ* location of phytoliths was deciphered. In the present study, silica percentage was studied from different parts of the plant and the maximum amount was found in the leaf. Frequency and morphometric data of phytolith morphotypes from different parts of the plants were also collected and analyzed. The strongest correlation was found between phytolith types of root and culm by Pearson's correlation coefficient supported by cluster analysis. The saddle type of phytoliths had the highest frequency in the leaf; other types of phytoliths in different parts of the plant were bilobate, blocky types, elongate types, trapezoids, triangular, cross, sinuate elongate, tabular types, globular types. Functional groups and amorphous polymorphic phases of silica were also analyzed by FTIR and XRD. It was concluded that phytolith types are controlled by parts of plant body and by anatomical and environmental factors.

## Introduction

The grass family, Poaceae comprises of 768 genera and 11506 grass species [1–4]. *Acrachne racemosa* (B. Heyne ex Roem. & Schult.) Ohwi of subfamily Chloridoideae (family Poaceae) is an annual weed of monsoon season and is commonly known as Chinkhe. It is widely distributed in Indian Punjab, Kashmir, N. W. F. P (Pakistan), Southeast Asia, tropical Africa, Ceylon, and northern Australia [5]. *A. racemosa* belongs to the tribe Cynodonteae that contains 21 subtribes, 94 genera and 850 species [4].

**Funding:** The author(s) received no specific funding for this work.

**Competing interests:** The authors have declared that no competing interests exist.

Grasses have been used as stimulants and laxatives, nerve tonic, and as a cure for diarrhea, dysentery, fever, cough, and jaundice [6, 7]. *A. racemosa* was first reported from Kanchanaburi province (Thailand) and it has now been reported as a weed in upland crops like cassava, maize, sunflower, roadsides and in lawns, cultivated fields, and waste places. The flowering and fruiting time is July-September. It is used in the treatment of piles and blood purification and as a diuretic. *A. racemosa* has the *in-vitro* antioxidant potential [5]. The grass species is an excellent fodder for cattle and the grain is rarely infected by insects.

Phytoliths are mainly made of Silicon which is the eighth-most abundant element in the universe and the second most abundant element in the earth's crust [8, 9]. It comprises 75% of the rock material [10]. Silicon contained in epidermal tissues, cell walls, and phytoliths ranges from 15 to 79% [11]. Phytoliths also contain a small amount of Al, Ca, Fe, K, Mg, Mn, Na, and organic carbon (C) [12–14] along with main components (SiO2 and H2O). Phytoliths are inorganic, resistant to decay, and are durable fossils of terrestrial plants. They are released into the soil after the decomposition of plants [10, 15–17]. They occur in all types of plants and their parts. Phytoliths help the plant to resist abiotic (physical and chemical) and biotic stresses, diseases, and pathogens [18–20] and help in the growth and development of plants [21] by providing strength, durability, and mechanical support to the culm. Physical stresses include drought, radiation, high and low temperature. There is variation in distribution, diversity, and frequency of opal phytoliths in the leaves, culm, root, and synflorescence [22].

Uptake of Silica (Si) is achieved by three processes namely active, passive, and rejective [23–25]. Diffusion is the process by which silica is absorbed by roots in the form of orthosilicic acid and it gets translocated via transpiration stream along with water [24]. Adsorption of silicic acid is pH-dependent [26]. Through root, silica is translocated to the shoot along with water. Silica is deposited in amorphous hydrated silica bodies ($SiO_2.nH_2O$) called phytoliths or silica bodies or plant opal [27] which are rigid microscopic structures that develop in inter and intra-cellular spaces of different organs namely culms, leaves, roots, and synflorescence, and may constitute up to 20% of the dry weight [22]. Phytoliths have been reported since the Late Devonian period [28]. Silica accumulates more in monocots and herbaceous species of families such as Poaceae, Cyperaceae, Commeliaceae, and Zingiberaceae. In grasses, it is present in sub-epidermal and epidermal layers of leaf and other parts including the protective covers of glumes and lemmas and the epidermal layers of the caryopsis [29–32]. The largest amount of Si is absorbed by sugarcane [33] (300–700 kg Si ha$^{-1}$), rice (150–300 kg Si ha$^{-1}$), wheat (50–150 kg Si ha$^{-1}$) [9].

Phytoliths are classified into two types (the cell wall type and the lumen type) depending on the place of deposition of silica [33–36] and they differ by plant genotypes. However, in grasses, lumen type of phytoliths dominate [33–35, 37–39]. Phytoliths' shapes depend upon the size and shape of the host cell [34, 35, 40, 41]. Phytoliths diagnose family, genus, and even arboreal or woody habits [42].

Types of phytoliths help in the characterization of tribes and genera within Chloridoideae [17]. Based on globular echinate types of phytolith a classification system was proposed which allowed the identification of tribes and subtribes of Amazonian palms [43]. The study of phytoliths helps in the identification and taxonomic demarcation of grass species and helps to distinguish C$_3$ from C$_4$ grasses [44–47]. Information on the phytolith profiles of the foxtail grass genus has been employed as signatures for taxonomic demarcation of species [48]. Phytoliths have helped to demarcate and identify some taxa and communities in the soil records of six community types in the Altay [49]. A study of bark phytoliths by [50] in African plants revealed several taxonomically useful morphotypes.

Phytoliths from plant fossils have given additional information about changes in the C3/C4 plant ratios providing new perspectives for the reconstruction of vegetation dynamics in the

northern temperate forest region and interpretation of the rapid changes in vegetation. The data correlates well with the patterns evident in the pollen data from stratigraphically equivalent locations [51]. [52] found that spheroidal phytoliths in the roots of the date palm look like echinates in the aerial parts. Six ecosystems were differentiated by soil phytolith assemblages in the ancient Maya tropical lowlands [53]. [54] found that soil micro aggregates which was the topsoil size fraction account for over 60 percent of the phytoliths and control bioavailable Si as phytoliths protect the element from rapid dissolution and release. Phytoliths gave evidence of the changes in regional climate [55].

The presence of silica skeleton and elongate dendritic phytolith has given evidence of crop processing on the Taraschina site and phytolith analysis conducted in the Danube Delta showed that Chalcolithic populations could grow cereal, ca. 6000 years ago in the heart of Delta [56]. The silica microfossil record which was taken from the sedimentary sequence of the Pleistocene/Holocene period from the Eastern Chaco Region showed that the most abundant microfossils were phytoliths [57]. In recent years, phytoliths have been used as palaeoenvironment, archaeological proxies and for plant-people relationships [42, 53, 57–61]. However, phytolith work is labor-intensive and expensive like most archaeobotanical and paleoenvironmental work [62].

The objective of this study was to develop a phytolith profile of *A. racemosa*, through a study of diversity, morphometry, and frequency data from different portions of the plant, namely, root, culm, leaf, and synflorescence. As per the latest records, no previous work has been done on the phytoliths of this species.

## Material and methods

### Area of study

*A. racemosa* was collected from the campus of Guru Nanak Dev University (Amritsar) which is spread over an area of about 500 acres at 31.31˚N and 74.55˚E. Species specimen has been submitted in the herbarium of Department of Botanical and environmental sciences, Guru Nanak Dev University, Amritsar with collection no. 7578 as shown in S1 Fig. This species is readily available from nature and it has no conservation issues. As such, no ethical questions were involved. Whole plant specimens were collected at the flowering stage along with the synflorescence. Specimens were separated into four fractions; culm, leaf, and synflorescence (above ground) and roots (underground) and preserved in 70% ethanol at 4˚C for *the in-situ* location of phytoliths. The remaining material was washed and then dried for Dry Ash extraction.

### *In-situ* location

The method of [63] with some modifications was used for studying the *in-situ* location of phytolith. Mature leaves were first cut into segments and boiled in distilled water for 5–10 minutes in test tubes (50 mL) which were put in ethanol (absolute) and heated (80˚C) in a water bath till green color was removed. Thereafter, in a solution of lactic acid and chloral hydrate (3:1 v/v), the segments were immersed and boiled again for 20–30 min in a water bath. The leaf segments were placed on clean ceramic tiles with the abaxial surface upwards and then the epidermis was peeled off the middle part of the leaf blades. In the same way, peelings from the adaxial surface of leaf segments were obtained. Thereafter, the peelings were stained in Gentian Violet and heated over in a watch glass over spirit lamb for deep staining and passed through a dehydration series of ethanol (30% through 50, 70, 90%, and absolute ethanol) and then mounted in DPX for light microscopy by Mag Cam DC—14 at a magnification of 40x.

## Extraction

For dry ashing of the plant material, standard protocol of [40] was employed with some modifications. The plant specimens were separated into different parts: root, culm, leaves, and synflorescence; washed and were dried in an oven. The dried parts were weighed and transferred to porcelain crucibles which were put into the muffle furnace and incinerated at 550 ˚C for 4–6 hours to ashes. Then, the ash contents were cooled, weighed, and transferred to test tubes which were incubated in 30% hydrogen peroxide at 80˚C for 1 hour in a preheated water bath until the material settled down. After that, the mixture was rinsed twice in distilled water and incubated in 10% HCl for 1 hour. The test tubes were then taken out, the mixture was washed in distilled water, centrifuged for 15 min at 7,500 rpm and the supernatant was decanted off. The pellet was washed in distilled water till the pellet was clear. Thereafter, the pellet was put in an oven and dried at 60˚C to powder form, weighed, and stored in an eppendorf.

The silica content (%) was calculated by the formula: ash content/dry mass ×100.

## Morphometry and distribution

Five slides for each sample were prepared by putting a drop of a mixture (a small amount (ca. 0.1 mg) of dried ash dipped in 10 mL of Gentian violet in a watch glass) on the slides, heated gently and the excess stain was drained off, covered with cover slip by putting DPX. Mag Cam DC—14 was used for photography of different morphotypes of phytoliths at a uniform magnification of 40X. The frequency of different types of phytoliths was calculated. Phytolith morphotypes were given names according to the standard system of the International Code of Phytolith Nomenclature (ICPN 1.0; [64]; ICPN 2.0; ICPT, [65]. Image J software (version 1.46r.) was used for morphometric measurements of morphotypes of phytolith. This software records dimensions and calculated values of size, surface area, and shape including aspect ratio, roundness, solidity. PAST (Paleontological Statistics) software was used to calculate mean and standard error was used as a measure of dispersion.

## Biochemical analysis

Fourier Transform Infrared Spectrophotometer (FTIR) Cary 630 was carried out at room temperature by standard KBr method. Infra-red spectra of silica powder produced by different parts of the *A. racemosa* were recorded, analysed and represented in graphical form with Origin pro 8.6 software. MiniFlex II X-ray diffractometer, Rigaku was used for determining the XRD pattern of phytoliths from different parts of the plant. Origin pro 8.5 was used for making graphs and PCPDF-WIN Software was used for data analysis.

# Results and discussion

*A. racemosa* has erect, simple or branched, tufted culms and glabrous compressed leaf sheath. Identification of this grass has been done by using online sources; The Online World grass flora [66], Grass Phylogeny Working Group [67], Grass Phylogeny Working Group II [68].

## Ash and silica concentration

*A. racemosa* deposits silica in the form of phytolith in different parts of the body. The dry Ashing method on different parts of the plant revealed the highest percentage of ash and silica accumulated in leaf (17.45% and 8.49%) followed by root (13.51% and 5.10%), synflorescence (9.67% and 3.27%), and culm (8.36% and 1.31%) as shown in Fig 1. Si content varies from 0.1 to 10% Si on a dry weight basis in plant shoots [24]. The present work showed that silica content gets accumulated more in leaves which conforms to earlier findings [48, 69–71]. The aerial

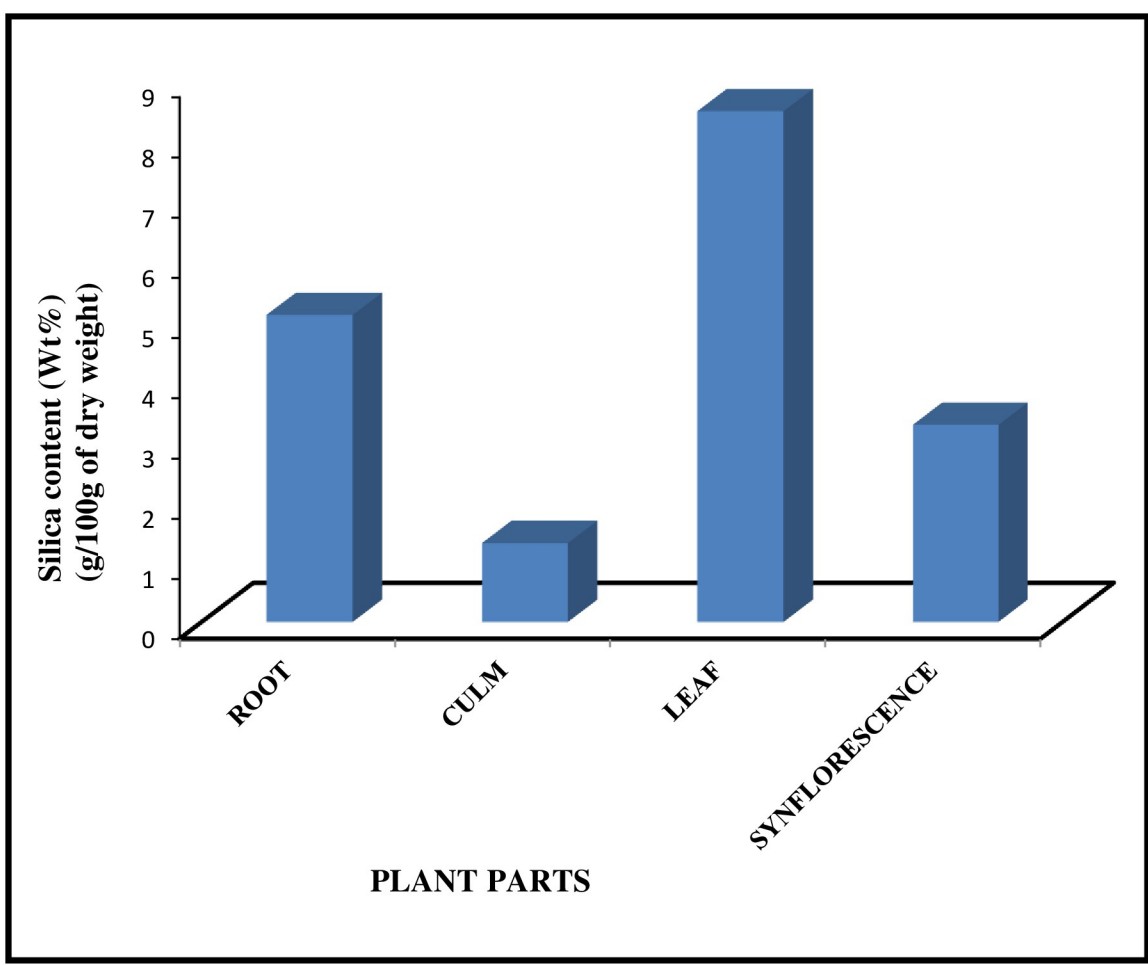

**Fig 1. 3D bar-chart showing silica content in various parts of *Acrachne racemosa* (Heyne ex Roem. & Schult.) Ohwi.**

parts of the plant body including epidermal cells, their associated structures, the cells of vascular bundles, and also the spaces between the cortical cells are believed to be the targeted sites of silicification [72, 73]. The higher levels of silicification in aerial plant parts such as leaf laminae and the synflorescence bracts have been correlated with higher evapotranspiration rates and once absorbed, silica is transported *via* the xylem to various plant parts through the transpiration stream. Silicic acid solutes are progressively concentrated during transpiration when water evaporates which results in super-saturated concentrations of Si(OH) and its deposition in tissues is in the form of amorphous silica i.e. phytoliths; the extent of supersaturation being controlled by the concentration of silicic acid in soil water [74–76]. It has been shown experimentally that deficiency of silicon in plants leads to excessive transpiration and its deposition is controlled by transpiration rate [19, 77] also proposed that the highest silica deposition occurs in the leaf blade in which most of the transpiration takes place.

### *In-situ* location (Epidermal peeling)

After cytology, leaf epidermis plays the most important role in the taxonomy of plant species [78–84]. The grass epidermis has three categories of elements namely bulliform elements, differentiated elements, fundamental elements [85]. *In-situ* location of phytoliths in the leaf

epidermal layer of *A. racemosa* revealed that phytoliths were presented in both adaxial and abaxial surfaces of the leaf. The short cells and the long cells are arranged in diagnostic patterns in the costal and intercostal regions of the leaf epidermis [86]. The adaxial surface had 1–10 rows of saddle phytoliths longitudinally arranged and separated by silica short cells in the costal region and 1–4 rows of stomata cells separated by long epidermal cells in the intercostal region (Fig 2A). The costal region is comprised of the acute bulbous, elongate echinate, saddle, and saddle with notch phytoliths (Fig 2A and Table 1). Elongate echinate with concave ends separated by stomatal cells were found in the intercostal region (Fig 2Ab). Short cells were found between long epidermal cells in the intercostal region as shown in Fig 2A. The abaxial surface showed 1–6 rows of saddle phytoliths arranged longitudinally in the costal region separated by short cells and 1–2 rows of stomatal cells separated by long epidermal cells in the intercostal region which were less frequent than the adaxial region (Fig 2B). The phytoliths in the abaxial surface had an acute bulbous, bilobate, cross, elongate echinate, ovate in the costal region and acute bulbous in the intercostal region (Fig 2B and Table 1). Saddle-shaped phytoliths are present in almost all the members of the subfamily chloridoideae [87]. In the present work also, saddle-shaped phytoliths are found most frequently in the epidermal layer of the leaf as *A. racemosa* is a member of tribe cyanodonteae which belong to subfamily Chloridoideae. They are the characteristics feature and dominant phytolith class of the Chloridoideae [32, 88]. [27] showed the presence of silica bodies on both adaxial and abaxial surfaces of leaf epidermis of Koeleria *macrantha*. Phytoliths give significant taxonomic information due to their consistent shape within species [46, 48, 71, 89–93]. [71] distinguished *Arundo donax* and *Phragmites karka* based on epidermal patterning of leaves [48]. Distinguish three species of the genus *Setaria* based on phytoliths pattern in the adaxial and abaxial surface of the leaf.

## Phytolith morphotypes

Phytoliths have been reported mainly from aboveground parts like culm, leaf, and synflorescences [17, 22, 85, 88, 94–104]. But in the present paper, we have increased the scope of phytolith analysis in Poaceae by the study of phytoliths from the underground parts (root) as well [105]. Studied phytoliths from native plants as well as the soil surface. Phytolith morphotypes play an important role in taxonomical demarcation as shown by three species of the foxtail genus which were classified based on phytolith morphotypes [48]. A total of 43 phytolith morphotypes were identified in the present paper according to their diagnostic shapes and sizes. Of the cumulative number of morphotypes, morphometry of 11 morphotypes has been done from the root (Table 2), 12 from culm (Table 3), 15 from leaves (Table 4) and 10 from synflorescence (Table 5). These morphotypes were divided into six groups namely; blocky types, short cells, long cells, bulliform cells, globular types, and tabular types. (Tables 2–5 and Figs 3–6). Blocky types of phytoliths originate from endodermis and the remaining morphotypes arise from the epidermis [86, 103].

## Underground part (Root)

Phytolith morphotypes have been well documented in the Poaceae family, except for the blocky and globular types [86, 101, 106], which are considered to be distinctive of forest trees [107]. In the roots of *A. racemosa*, twenty-six (26) morphotypes were present including mostly blocky, globular, tabular, trapezoid (Fig 3l), triangular (Fig 3g–3k), cuneiform bulliform (Fig 3x and 3y). Blocky morphotypes include blocky irregular (Fig 3u), blocky polyhedral (Fig 3d and 3s) (Table 2). The blocky types morphotypes are found in this study have previously been reported in some grasses [34, 108]. Globular morphotypes include globular granulate (Fig 3b and 3c), globular polyhedral (Fig 3e and 3f), globular psilate (Fig 3a) and tabular types include

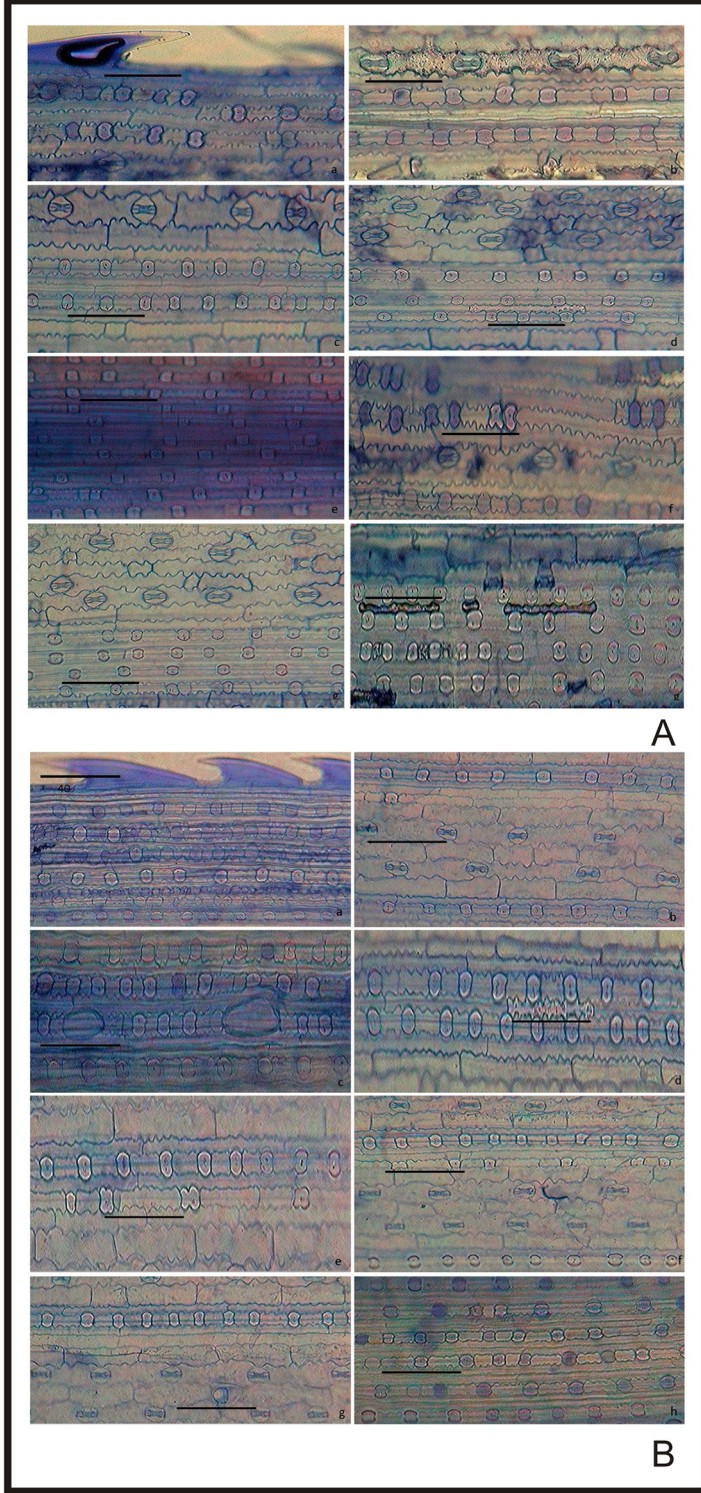

**Fig 2.** *In-situ* **location of phytoliths in leaf epidermis of *Acrachne racemosa* (Heyne ex Roem. & Schult.) Ohwi.** A. Adaxial surface (a-h) B. Abaxial surface(a-h); bar = 40 μm.

**Table 1. Type of phytolith morphotypes in costal and intercostal region of adaxial and abaxial leaf surface of *Acrachne racemosa* (Heyne ex Roem. & Schult.) Ohwi.**

| S.No. | Leaf surface | Region | Phytolith morphotypes |
|---|---|---|---|
| 1. | Adaxial | Costal | 1–10 rows of saddle phytoliths |
| | | | Acute Bulbous |
| | | | Elongate echinate |
| | | | Saddle |
| | | | Saddle with notch |
| | | Intercostal | 1–4 rows of stomatal cells |
| | | | Elongate echinate with concave ends |
| 2. | Abaxial | Costal | 1–6 rows of saddle phytolith |
| | | | Acute Bulbous |
| | | | Bilobate |
| | | | Cross |
| | | | Elongate echinate |
| | | | Ovate |
| | | Intercostal | 1–2 rows of stomatal cells |
| | | | Acute Bulbous |

tabular simple (Fig 3m), tabular irregular (Fig 3v), and tabular polyhedral (Fig 3t) (Table 2). Acute bulbous (Fig 3p) and prism (Fig 3q and 3r) were the morphotypes that were less frequent.

## Vegetative parts (Culm and leaves)

Twenty-eight (28) morphotypes were recovered from the vegetative part (culm). The morphotypes that were present include acute bulbous (Fig 4u), amoeboid (Fig 4ah), blocky types, cavate (Fig 4s), cuboid (Fig 4ac and 4ad), cuneiform (Fig 4t), cuneiform bulliform (Fig 4o–4r), elongate types, globular of varied types, horned tower (Fig 4w and 4x), long trapezoid (Fig 4y and 4ab), nodular (Fig 4ag), ovate (Fig 4i), pentagon (Fig 4k and 4l), polyhedral (Fig 4j and 4m), stellate (Fig 4n), tabular simple (Fig 4z), rtrapezoid (Fig 4v and 4aa) and triangular (Fig 4h). Globular types include globular psilate (Fig 4a and 4b), globular granular (Fig 4c and 4d),

**Table 2. Morphometry of phytolith morphotypes of *Acrachne racemosa* (Heyne ex Roem. & Schult.) Ohwi (Root).**

| Phytolith morphotypes | Length(μm) | Width(μm) | Area(μm$^2$) | Perimeter(μm) | Aspect ratio(μm) | Round | Solidity |
|---|---|---|---|---|---|---|---|
| Blocky irregular (Fig 3u) | 379.343±70.460[#] | 233.442±15.858 | 81314.39±19670.18 | 1140.244±165.658 | 1.520±0.140 | 0.674±0.059 | 0.937±0.004 |
| Blocky polyhedral (Fig 3d and 3s) | 402.748±72.924 | 343.178±75.985 | 126498.5±49729.32 | 1309.179±234.646 | 1.204±0.086 | 0.845±0.050 | 0.941±0.006 |
| Cuneiform bulliform (Fig 3x and 3y) | 288.182±43.156 | 171.654±27.003 | 41012.55±8661.031 | 817.675±99.196 | 1.688±0.201 | 0.632±0.084 | 0.929±0.016 |
| Globular granulate (Fig 3b and 3c) | 190.610±34.975 | 156.807±30.891 | 26544.26±9373.74 | 586.945±101.186 | 1.224±0.053 | 0.822±0.034 | 0.965±0.005 |
| Globular polyhedral (Fig 3e and 3f) | 369.901±49.715 | 324.382±40.243 | 104258.8±26316.2 | 1198.609±150.028 | 1.179±0.047 | 0.853±0.034 | 0.961±0.002 |
| Globular psilate (Fig 3a) | 226.104±68.549 | 180.571±36.610 | 46335.78±17029.58 | 857.482±183.119 | 1.342±0.152 | 0.778±0.074 | 6.902±0.015 |
| Tabular simple (Fig 3m) | 319.472±28.303 | 213.194±32.499 | 57094.5±12768.77 | 956.717±86.895 | 1.783±0.345 | 0.598±0.097 | 0.975±0.003 |
| Tabular irregular (Fig 3v) | 282.372±11.527 | 170.795±23.987 | 40260.47±4905.078 | 849.67±68.127 | 1.621±0.274 | 0.660±0.130 | 0.93±0.019 |
| Tabular polyhedral (Fig 3t) | 361.993±54.280 | 233.725±22.191 | 67548.51±11089.22 | 1083.532±93.538 | 1.381±0.179 | 0.751±0.105 | 0.920±0.018 |
| Trapezoid (Fig 3n and 3o) | 226.104±68.549 | 180.571±36.601 | 46335.78±17029.58 | 857.482±183.119 | 1.342±0.152 | 0.778±0.074 | 0.902±0.015 |
| Triangular (Fig 3g–3k) | 227.623±5.833 | 171.053±15.528 | 36756.63±5520.581 | 797.595±81.609 | 1.314±0.127 | 0.788±0.071 | 0.093±0.026 |

[#] = mean±Standard Error.

**Table 3. Morphometry of phytolith morphotypes of *Acrachne racemosa* (Heyne ex Roem. & Schult.) Ohwi (Culm).**

| Phytolith morphotypes | Length(μm) | Width(μm) | Area(μm²) | Perimeter(μm) | Aspect ratio(μm) | Round | Solidity |
|---|---|---|---|---|---|---|---|
| Amoeboid (Fig 4ah) | 589.708±55.912[#] | 303.736±23.261 | 145719.3±16786.71 | 2051.349±151.842 | 1.766±0.203 | 0.601±0.078 | 0.810±0.028 |
| Blocky irregular (Fig 4ai) | 398.055±33.597 | 256.231±21.489 | 84656.54±10361.61 | 1335.117±169.238 | 1.6328±0.215 | 0.650±0.074 | 0.897±0.034 |
| Blocky polyhedral (Fig 4aj) | 438.213±113.984 | 321.924±87.034 | 149452.8±66586.71 | 1391.069±401.813 | 1.366±0.093 | 0.744±0.044 | 0.905±0.022 |
| Cuneiform bulliform (Fig 4o–4r) | 163.513±16.872 | 96.628±11.871 | 14105.36±1816.47 | 526.007±32.829 | 1.415±0.122 | 0.728±0.064 | 0.869±0.008 |
| Globular granulate (Fig 4c and 4d) | 155.218±53.456 | 192.501±60.293 | 33537.57±21492.96 | 635.614±208.892 | 1.221±0.090 | 0.832±0.060 | 0.951±0.006 |
| Globular polyhedral (Fig 4g) | 160.229±23.446 | 121.046±17.449 | 15545.01±3678.835 | 484.837±59.954 | 1.438±0.097 | 0.704±0.044 | 0.953±0.011 |
| Globular echinate (Fig 4e and 4f) | 240.331±16.582 | 204.532±13.714 | 43134.8±5028.326 | 792.066±41.644 | 1.140±0.004 | 0.877±0.003 | 0.968±0.003 |
| Nodular (Fig 4ag) | 615.903±90.236 | 251.554±18.375 | 121745.8±27855.08 | 1814.402±294.774 | 2.430±0.279 | 0.437±0.058 | 0.831±0.011 |
| Ovate (Fig 4i) | 161.679±61.404 | 92.008±18.967 | 15523.11±9336.89 | 461.313±161.987 | 1.673±0.208 | 0.617±0.081 | 0.963±0.005 |
| Pentagon (Fig 4k and 4l) | 172.156±32.332 | 118.469±25.116 | 19122.98±6665.892 | 512.345±94.098 | 1.568±0.234 | 0.685±0.083 | 0.956±0.008 |
| Trapezoid (Fig 4v) | 124.561±34.594 | 70.781±11.188 | 10419.71±4775.453 | 395.057±92.285 | 1.592±0.215 | 0.670±0.081 | 0.935±0.013 |
| Triangular (Fig 4h) | 255.542±76.860 | 162.254±48.647 | 43906.77±22406.43 | 827.952±279.091 | 1.522±0.088 | 0.667±0.043 | 0.926±0.011 |

[#] = mean±Standard Error.

**Table 4. Morphometry of phytolith morphotypes of *Acrachne racemosa* (Heyne ex Roem. & Schult.) Ohwi (Leaves).**

| Phytolith morphotypes | Length(μm) | Width(μm) | Area(μm²) | Perimeter(μm) | Aspect ratio(μm) | Round | Solidity |
|---|---|---|---|---|---|---|---|
| Acute Bulbous (Fig 5aaw) | 378.957±25.391[#] | 119.415±7.428 | 43856.43 ±2885.113 | 1160.483±72.710 | 2.863±0.390 | 0.378 ±0.053 | 0.824 ±0.028 |
| Bilobate (Fig 5r–5u) | 86.796±7.577 | 40.388±5.072 | 4365.505±499.403 | 281.597±17.948 | 1.800±0.094 | 0.562 ±0.030 | 0.903 ±0.007 |
| Cuneiform bulliform (Fig 5aq–5ar) | 214.037±23.049 | 153.705 ±20.898 | 28528.75 ±5352.749 | 657.435±70.862 | 1.362±0.125 | 0.759 ±0.068 | 0.938 ±0.007 |
| Cross (Fig 5n and 5o) | 68.724±8.620 | 36.653±2.519 | 3340.076±625.009 | 265.748±24.034 | 1.515±0.155 | 0.682 ±0.056 | 0.824 ±0.015 |
| Elongate echinate (Fig 5ax and 5aaq–5aav) | 251.542±28.065 | 61.961±12.994 | 14604.16 ±4411.486 | 646.804±73.411 | 4.686±0.620 | 0.228 ±0.029 | 0.814 ±0.025 |
| Globular granulate (Fig 5ae–5af) | 134.595±10.933 | 121.762 ±12.008 | 13784.04 ±1749.663 | 444.86±28.536 | 1.098±0.038 | 0.912 ±0.032 | 0.963 ±0.002 |
| Globular polyhedral (Fig 5ag) | 268.507±49.171 | 227.241 ±32.153 | 51808.01 ±14583.03 | 856.877±127.260 | 1.171±0.055 | 0.861 ±0.039 | 0.949 ±0.004 |
| Ovate (Fig 5y) | 201.821±39.412 | 85.719±10.605 | 14873.88 ±4576.796 | 501.731±79.297 | 2.201±0.200 | 0.462 ±0.643 | 0.971 ±0.001 |
| Parallelepipedal cell (Fig 5an) | 234.032±18.687 | 162.536 ±13.025 | 34420.46 ±3931.019 | 743.445±43.405 | 1.537±0.184 | 0.687 ±0.077 | 0.963 ±0.006 |
| Polylobate (Fig 5v) | 175.925±12.401 | 59.907±3.967 | 8923.322±380.834 | 482.932±10.400 | 3.088±0.564 | 0.343 ±0.053 | 0.863 ±0.009 |
| Saddle (Fig 5a–5k) | 110.828±20.077 | 106.408 ±21.303 | 16920.86 ±5272.289 | 513.573±97.255 | 1.084±0.021 | 0.923 ±0.017 | 0.929 ±0.004 |
| Sinuate elongate (Fig 5aae) | 652.793 ±134.616 | 86.639±14.851 | 54075.21 ±22211.24 | 1476.7±284.617 | 7.461±1.128 | 0.145 ±0.019 | 0.812 ±0.051 |
| Smooth elongate (Fig 5aaa and 5aaj–5aan) | 476.720±60.180 | 49.558±3.616 | 23826.71 ±4035.366 | 1056.635 ±121.187 | 9.467±1.460 | 0.114 ±0.014 | 0.876 ±0.006 |
| Tabular simple (Fig 5ak and 5al) | 291.878±78.062 | 124.564 ±16.808 | 37172.43 ±10470.71 | 828.835±164.706 | 2.318±0.573 | 0.478 ±0.094 | 0.954 ±0.001 |
| Trapezoid (Fig 5ap) | 153.488 ±20.979 | 136.633 ±22.150 | 18562.37 ±2518.617 | 593.912±41.263 | 1.7±0.230 | 0.608 ±0.072 | 0.913 ±0.016 |

[#] = mean±Standard Error.

**Table 5. Morphometry of phytolith morphotypes of *Acrachne racemosa* (Heyne ex Roem. & Schult.) Ohwi (Synflorescence).**

| Phytolith morphotypes | Length(µm) | Width(µm) | Area(µm$^2$) | Perimeter(µm) | Aspect ratio(µm) | Round | Solidity |
|---|---|---|---|---|---|---|---|
| Acute Bulbous (Fig 6t–6v) | 475.017±89.370[#] | 138.412±12.883 | 40549.75±12277.54 | 1131.247±191.953 | 3.745±0.275 | 0.272±0.019 | 0.791±0.006 |
| Acicular (Fig 6y) | 251.436±42.469 | 42.618±2.676 | 7807.947±1713.322 | 559.121±94.103 | 5.283 ±0.798 | 0.207±0.031 | 0.888±0.013 |
| Bilobate (Fig 7a) | 67.441±7.934 | 26.144±0.900 | 1925.924±352.913 | 198.546±21.364 | 2.586±0.179 | 0.393±0.025 | 0.861±0.015 |
| Blocky irregular (Fig 6r and 6s) | 424.939±49.466 | 305.5258±48.364 | 118380.5±27312.05 | 1330.752±147.855 | 1.376±0.166 | 0.760 ±0.070 | 0.948±0.010 |
| Blocky polyhedral (Fig 6l and 6q) | 389.202±110.292 | 269.9042±44.581 | 86575.67±37950.88 | 1151.341±251.176 | 1.537±0.203 | 0.693±0.081 | 0.912±0.018 |
| Elongate echinate (Fig 6aa) | 451.322±69.663 | 44.520±8.682 | 20501.01±4180.472 | 1147.957±155.053 | 8.230±0.849 | 0.127±0.015 | 0.7148±0.013 |
| Globular granulate (Fig 6b and 6c) | 131.973±20.972 | 107.203 ±20.351 | 12644.06±3751.268 | 412.983±67.857 | 1.220±0.091 | 0.836±0.057 | 0.953±0.007 |
| Saddle (Fig 7f) | 57.175±2.500 | 48.636±2.588 | 2826.613±139.493 | 216.735±5.61233 | 1.212±0.101 | 0.844±0.059 | 0.9302±0.010 |
| Smooth elongate (Fig 6z) | 721.449±205.067 | 84.544±13.016 | 63306.26±18802.41 | 1621.715±415.699 | 8.483±2.392 | 0.152±0.031 | 0.9±0.023 |
| Trapezoid (Fig 6j) | 206.339±67.491 | 79.592±18.074 | 21280.83±10294.77 | 618.607±164.635 | 2.291±0.445 | 0.5172±0.106 | 0.8976±0.0193 |

[#] = mean±Standard Error.

globular echinate (Fig 4e and 4f), and globular polyhedral (Fig 4g) having variation in their frequencies. The Arecaceae (palm) family produces globular echinate phytoliths in abundance in every plant organ, and they were found in every non-Bactridineae species [105]. Blocky types include blocky irregular (Fig 4ai) and blocky polyhedral (Fig 4aj) and elongate type having smooth elongate (Fig 4ae) and elongate irregular (Fig 4af) of different frequencies. The most frequent morphotypes were globular granulate, globular echinate, globular polyhedral, triangular, ovate, pentagon, Cuneiform bulliform, trapezoid, nodular, amoeboid, blocky irregular, blocky polyhedral (Table 3) while other morphotypes were less frequent. Bulliform phytoliths of the Chloridoideae family are effective C4 plant indicators [109].

In the leaves, thirty-seven (37) morphotypes were studied which includes saddle (Fig 5a–5k), saddle with the notch (Fig 5l and 5m), cross (Fig 5n and 5o), flat tower (Fig 5p and 5q), bilobate (Fig 5r–5u), polylobate (Fig 5v), silica cork cells (Fig 5w), reniform (Fig 5x), ovate (Fig 5y), oblong (Fig 5z), ellipsoidal (Fig 5aa–5ad), blocky polyhedral (Fig 5ah), blocky (Fig 5aac–5aad), globular types, elongate types, tabular types, triangular (Fig 5ai), polyhedral (Fig 5aj), parallelepipedal cell (Fig 5an), cuboid (Fig 5ao and 5aab), trapezoids (Fig 5ap), cuneiform bulliform (Fig 5aq and 5ar), scutiform (Fig 5as–5au and 5ay), crescent moon (Fig 5av), rectangular (Fig 5aw), arcuate (Fig 5aap) and acute bulbous (Fig 5aaw). According to [103], short saddles are present only in *Distichlis spicata* but are also found in this study. Saddles are most common in xeromesophytes by [49] and were also reported in *Dactyloctenium aegyptium* (L.) and *Sporobolus diandrus* (Retz.) P. Beauv. by [110]. The saddle, bilobate, polylobate, cross, and bulliform phytolith morphotypes of grasses were also recovered from lake Malawi sediments by [61]. [111] found that bilobate type phytoliths had a lot of morphological diversity. *Sporobolus indicus* and *Eragrostis lugens*, as well as *Sporobolus piliferus*, have crescent, horned, and flat towers [87]. Elongate type of morphotypes consist of sinuate elongate (Fig 5aae), smooth elongate (Fig 5aaa and 5aaj–5aan), elongate irregular (Fig 5az and 5aao), elongate echinate (Fig 5ax and 5aaq–5aav), elongate/saddle (Fig 5aaf), and elongate echinate/saddle (Fig 5aag–5aai) in variable frequencies. Globular types of phytoliths include globular granulate (Fig 5ae and 5af), globular polyhedral (Fig 5ag), and tabular consist of tabular simple (Fig 5ak and 5al), tabular polyhedral (Fig 5am). Among these saddle, cross, polylobate, ovate, globular granulate, globular polyhedral, tabular simple, trapezoids, elongate echinate, sinuate elongate, smooth elongate, acute bulbous morphotypes were most frequent (Table 4). The bilobate, saddle-type phytoliths were also reported by [22] in leaf sheath, leaf blade, and culm of *Cynodon dactylon* (L.) Pers. by Laser-Induced Breakdown Spectroscopy (LIBS).

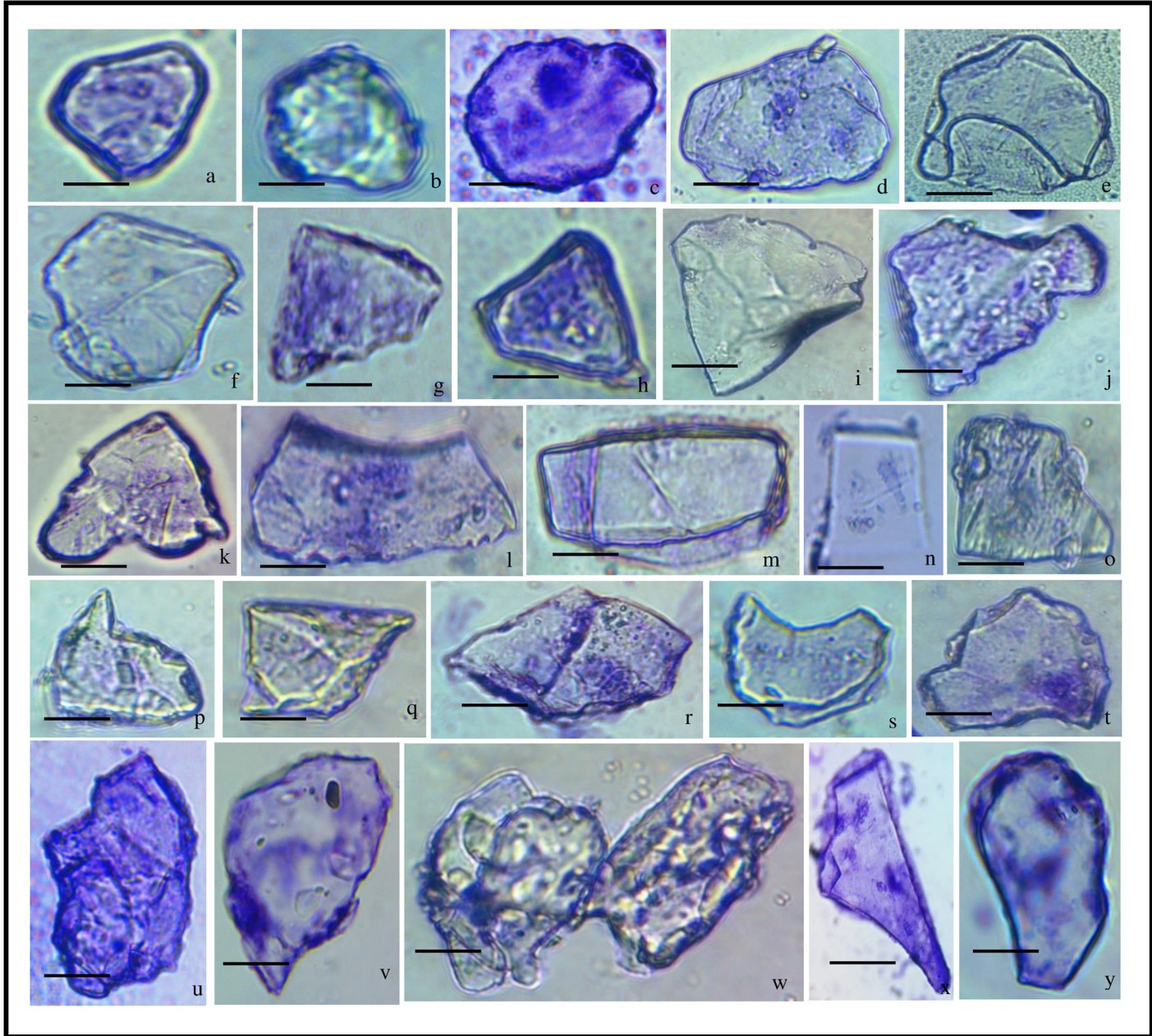

**Fig 3. Phytolith morphotypes of *Acrachne racemosa* (Heyne ex Roem. & Schult.) Ohwi (Root).** Globular psilate (a); Globular granulate (b, c); Blocky polyhedral (d); Globular polyhedral (e, f); Triangular (g-k); Trapezoid (l); Tabular simple (m); Trapezoid (n, o); Acute bulbous (p); Prism (q, r); Blocky polyhedral (s); Tabular polyhedral (t); Blocky irregular (u); Tabular irregular (v); Nodular (w); Cuneiform bulliform (x, y) bar = 20 μm.

## Fertile part (Synflorescence)

Twenty three (23) phytolith morphotypes along with undulation patterns complexed with other types of morphotypes were found in the synflorescence. Acute bulbous (Fig 6t–6v), acicular (Fig 6y), blocky irregular (Fig 6r and 6s), blocky polyhedral (Fig 6l and 6q), elongate echinate (Fig 6aa), globular granulate (Fig 6b and 6c), smooth elongate (Fig 6z), trapezoid (Fig 6j) (Table 5), globular psilate (Fig 6a), globular granulate (Fig 6b and 6c), triangular (Fig 6d and

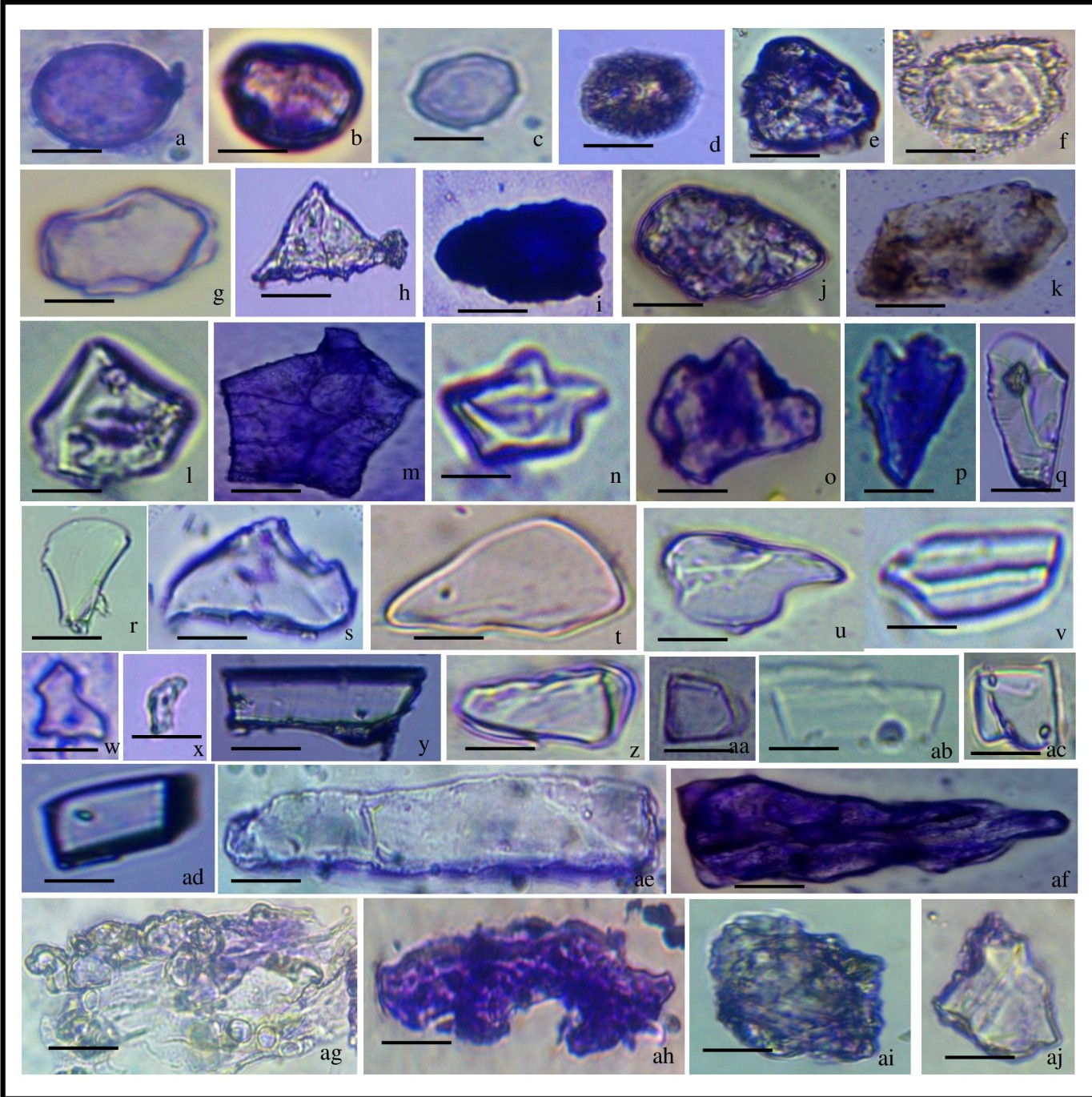

**Fig 4. Phytolith morphotypes of *Acrachne racemosa* (Heyne ex Roem. & Schult.) Ohwi (Culm).** Globular psilate (a, b); Globular granular (c, d); Globular echinate (e, f); Globular polyhedral (g); Triangular (h); Ovate (i); Polyhedral (j); Pentagon (k, l); Polyhedral (m); Stellate (n); Cuneiform bulliform (o-r); Cavate (s); Cuneiform (t); Acute bulbous (u); Trapezoid (v); Horned tower (w, x), Long trapezoid (y, ab); Tabular simple (z); Trapezoid (aa); Cuboid (ac, ad); Smooth elongate (ae); Elongate irregular (af); Nodular (ag); Amoeboid (ah); Blocky irregular (ai); Blocky polyhedral (aj) bar = 20 μm.

6e), cavate (Fig 6f), tabular simple (Fig 6g and 6h), long trapezoids (Fig 6i), trapezoid (Fig 6j), polyhedral (Fig 6k, 6m and 6n), tabular polyhedral (Fig 6o and 6p), and acute (Fig 6w and 6x) were the morphotypes reported in this part. Trapezoids are also reported from the inflorescence of *Pennisetum* by [111].

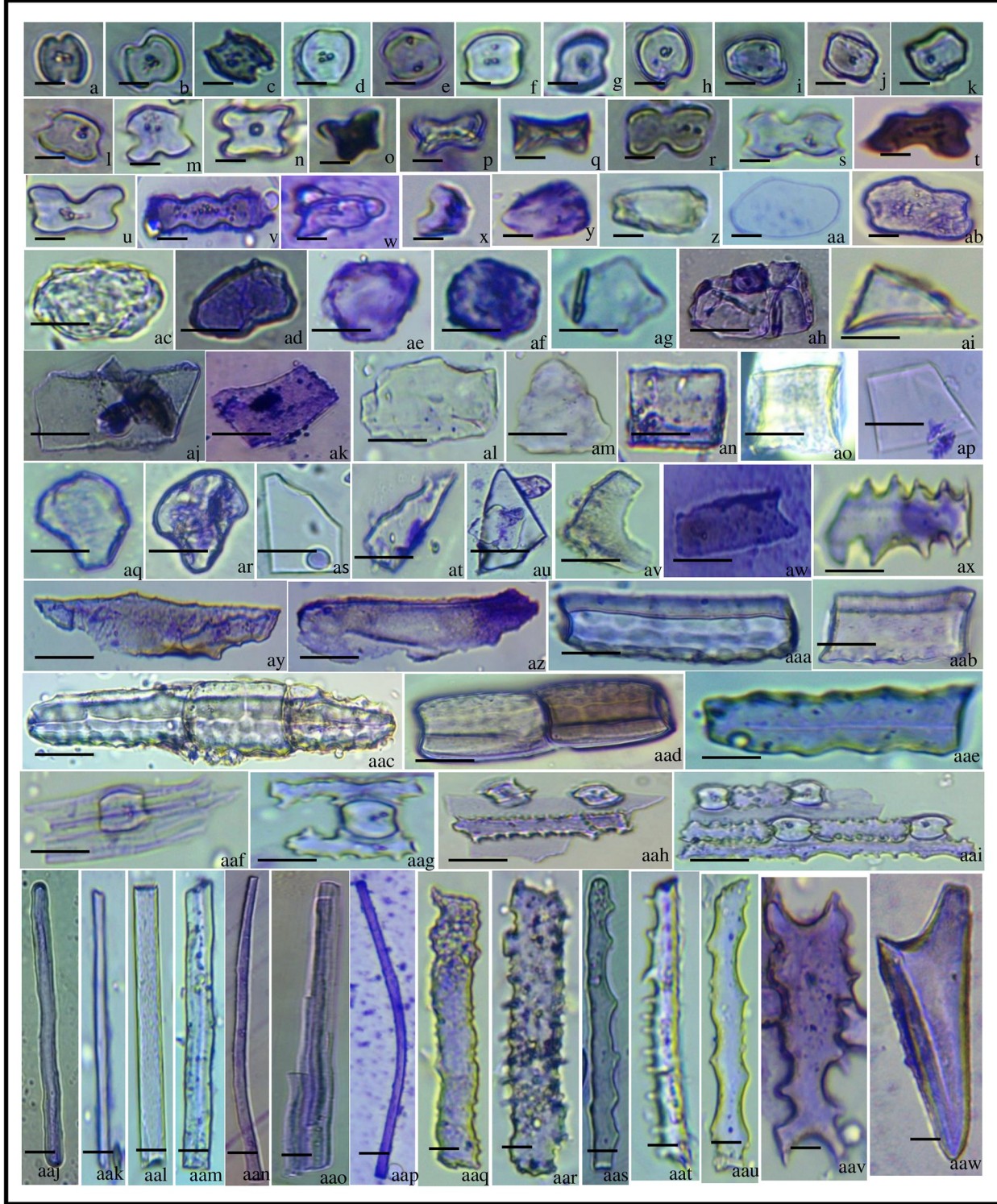

**Fig 5. Phytolith morphotypes of *Acrachne racemosa* (Heyne ex Roem. & Schult.) Ohwi (Leaves).** Saddle (a-k); Saddle with notch (l, m); Cross (n, o); Flat tower (p, q); Bilobate (r-u); Polylobate (v); Silica cork cells (w); Reniform (x); Ovate (y); Oblong (z); Ellipsoidal (aa-ad); Globular granulate (ae, af); Globular polyhedral (ag); Blocky polyhedral (ah); Triangular (ai); Polyhedral (aj); Tabular simple (ak, al); Tabular polyhedral (am); Parallelepipedal cell (an); Cuboid (ao); Trapezoids (ap); Cuneiform bulliform (aq, ar); Scutiform (as-au); Crescent moon (av); Rectangular (aw); Elongate echinate (ax); Scutiform (ay); Elongate irregular (az); Smooth elongate (aaa); cuboid (aab); Blocky (aac, aad); Sinuate elongate (aae); Elongate/Saddle (aaf); Elongate echinate/Saddle (aag-aai); Smooth elongate (aaj-aan); Elongate irregular (aao); Arcuate (aap); Elongate echinate (aaq-aav); Acute Bulbous (aaw). bar = 10 µm (a-ab, aaj-aaw); 40 µm (ac-aai).

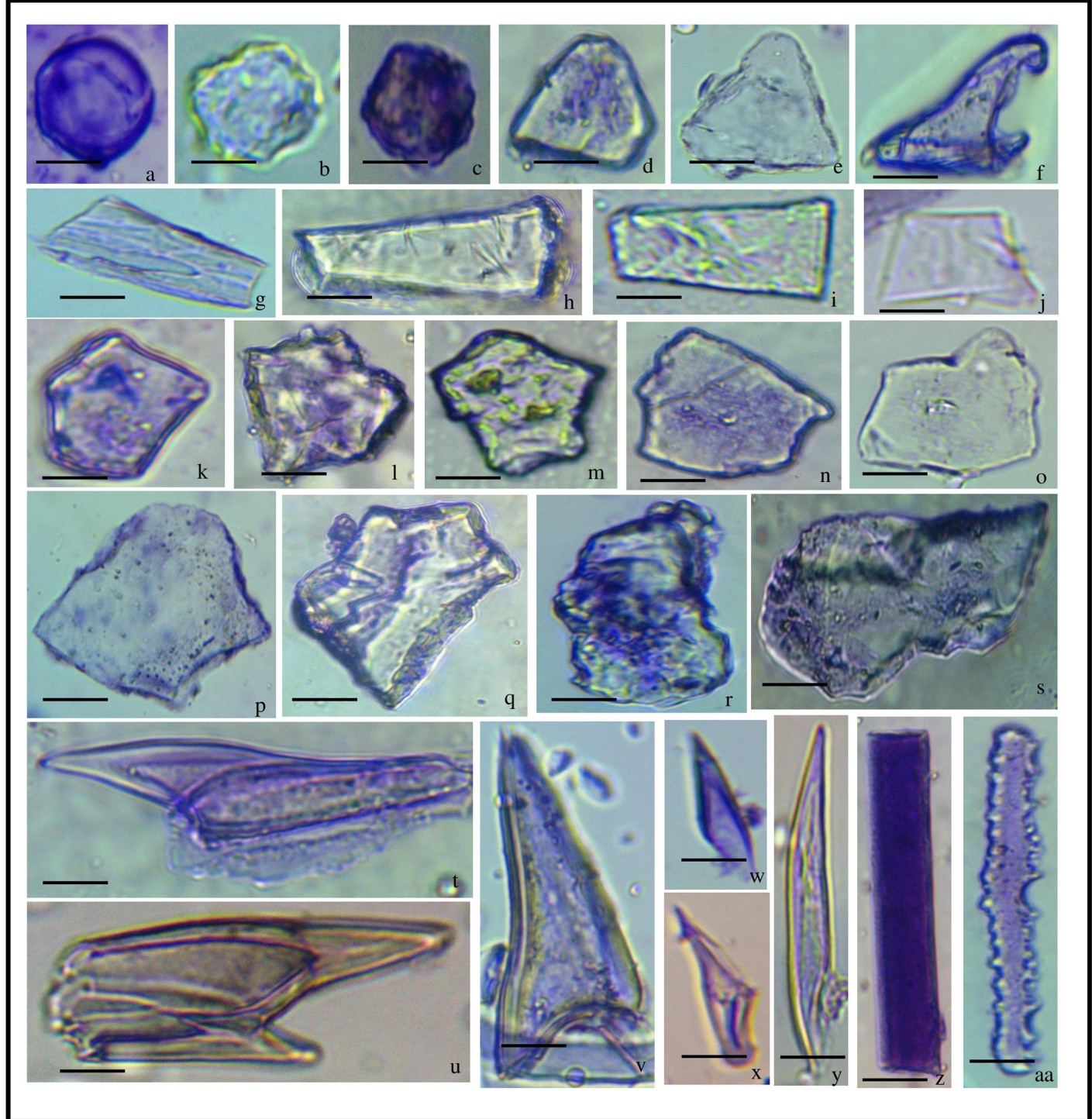

**Fig 6. Phytolith morphotypes of *Acrachne racemosa*(Heyne ex Roem. & Schult.) Ohwi (Synflorescence).** Globular psilate (a); Globular granulate (b, c); Triangular (d, e); Cavate (f); Tabular simple (g, h); Long trapezoids (i); Trapezoid (j); Polyhedral (k); Blocky polyhedral (l); Polyhedral (m, n); Tabular polyhedral (o, p); Blocky polyhedral (q); Blocky irregular (r, s); Acute bulbous (t-v) Acute (w, x) Acicular (y); Smooth elongate (z); Elongate echinate (aa): bar = 20 µm.

## Undulation patterns

The undulated patterns found in the synfloresence have been compared with the earlier findings [48, 112, 113]. We have taken the entire fertile part (synflorescence) from which these undulation patterns were studied (Fig 7). It showed elongate granulate and bilobate phytoliths were embedded in it (Fig 7a), It also shows elongate collumellate extensions on the epidermal long cells with embedded bilobates (Fig 7b), with embedded cork cells (Fig 7c). Columellate extensions of epidermal cells have also been reported in *Setaria pumila* [48]. With embedded prickle hair and bilobates, elongate granulate type of undulation was also found (Fig 7d and 7e). It also shows the presence of an elongate granulate type of margin on the epidermal long cell along with saddles (Fig 7f), cross (Fig 7h). Elongate clavate along with bilobates were also seen on the epidermal long cell of synflorescence (Fig 7g). Elongate castellate types of ornamentation were also seen (Fig 7j) along with saddle (Fig 7i) [114]. Classified undulation according to the type of undulation pattern, main body, and connection. Due to the differences in the undulation amplitude, there are differences in the morphology of the undulated patterns among Ω type, η type, and β type, and three important parameters were used to characterize the morphological variations of the structures of the β-type undulated patterns [112, 115].

## Frequency distribution and morphometric measurements

The data on frequency distribution and morphometric measurements of phytolith morphotypes have been employed as additional parameters for the identification of species [48, 71, 90, 95, 98, 105, 116]. The frequency of morphotypes shows variations in different parts of the plant species. In the underground part (root), blocky polyhedral has the highest frequency (22.30%) followed by blocky irregular (16.97%), globular granulate (9.39%), globular polyhedral (6.45%) whereas acute bulbous, cuboid, and prism present in low frequencies. Nodular phytolith present in roots and culm with 0.70% and 2.60% frequency. Mostly blocky types of phytoliths are found in roots (Fig 3). Blocky irregular (14.34%) constitute the highest frequency in the culm part followed by blocky polyhedral (9.56%), triangular (7.39%), globular granular (6.95%) whereas tabular simple, scutiform, stellate possess low frequency. Stellate type of phytolith found only in the culm part with the absence of acicular, oblong, flat tower types. In the vegetative part (leaf), saddle phytoliths were the most common category accounting for more than 50% of the total phytoliths and constitutes the highest frequency in leaf part followed by smooth elongate (8.84%), elongate echinate (6.63%), and bilobate (3.93%) and a very low frequency of reniform (0.04%), globular psilate (0.09%) with the absence of tabular irregular, amoeboid, stellate type of phytoliths. Saddle-shaped phytoliths are known to be diagnostic for members of subfamily Chloridoideae [32, 87, 108]. Saddle phytoliths were also present in synflorescence with very less frequency (2.77%) but are absent in the root and culm part of the species. The frequency of acute bulbous morphotype is highest in synflorescence (28.71%) followed by leaf (1.17%), culm (0.86%), and roots (0.42%). There is the absence of cuneiform bulliform, rectangular, cuboid morphotypes in the fertile part but are found in other parts of *A. racemosa*. Comparison of shared phytoliths in different parts i.e root, culm, leaves, and synflorescence also show significant differences. The frequency of smooth elongate is very less in the root (0.70%) as compared to culm (3.47%), leaf (8.84%), and synflorescence (7.05%). The frequency of trapezoid morphotypes was lesser in the leaf as compared to the root, culm, and synflorescence. Globular granular phytolith had a higher frequency in the root (9.39%) as compared to culm (6.95%), synflorescence (5.03%), and leaf (2.70%). Blocky irregular contributes the highest frequency in roots than in culm. Blocky polyhedral also contributes the highest frequency in roots (22.30%) as compared to culm (9.56%). Triangular phytoliths have the maximum frequency in culm then root, leaf, and synflorescence (Table 6).

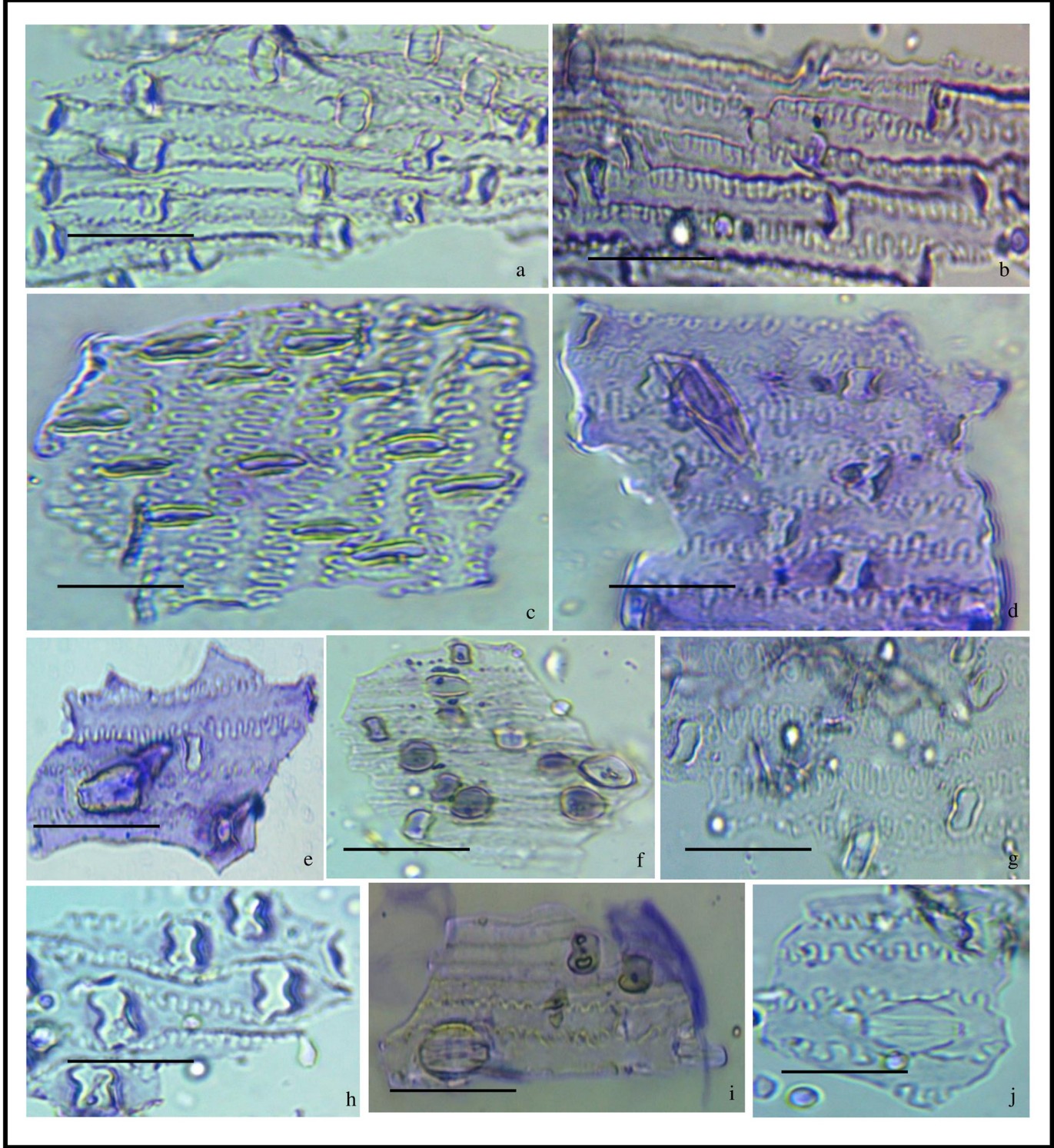

**Fig 7. Undulated patterns in *Acrachne racemosa*(Heyne ex Roem. & Schult.) Ohwi (Synflorescence).** Elongate granulate/bilobates (a); Elongate columellate/biolobate (b); Elongate Columellate/cork cells (c); Elongate granulate/bilobates/prickle hair (d, e); Elongate granulate/saddle (f); Elongate clavate/bilobates (g); Elongate granulate/cross (h); Elongate castellate/stomata/saddle (i); Elongate castellate (j). bar = 40 μm.

**Table 6. Frequency distribution of phytolith morphotypes from different parts of the *Acrachne racemosa* (Heyne ex Roem. & Schult.) Ohwi.**

| S.No. | Phytolith morphotype | Root (%) | Culm (%) | Leaf (%) | Synflorescence (%) |
|-------|---------------------|----------|----------|----------|--------------------|
| 1. | Acicular | 0.28 | - - - | 0.29 | 3.52 |
| 2. | Acute bulbous | 0.42 | 0.86 | 1.17 | 28.71 |
| 3. | Arcuate | - - - | - - - | 0.04 | 2.01 |
| 4. | Amoeboid | 0.98 | 4.34 | - - - | - - - |
| 5. | Bilobate | - - - | - - - - | 3.93 | 2.51 |
| 6. | Blocky irregular | 16.97 | 14.34 | 0.63 | 4.03 |
| 7. | Blocky polyhedral | 22.30 | 9.56 | 0.24 | 3.77 |
| 8. | Clavate | 0.42 | - - - | 0.14 | - - - |
| 9. | Crescent moon | - - - | - - - | 0.04 | - - - |
| 10. | Cross | - - - | - - - | 1.03 | - - - |
| 11. | Cuboid | 1.40 | 2.60 | 0.44 | - - - - |
| 12. | Cuneiform | 0.56 | 2.60 | 0.14 | 0.50 |
| 13. | Cuneiform bulliform | 3.92 | 4.34 | 0.09 | - - - |
| 14. | Ellepsoidal | 1.12 | - - - | 0.14 | - - - |
| 15. | Elongate echinate | - - - | 1.30 | 6.63 | 13.09 |
| 16. | Elongate irregular | - - - | 1.30 | 0.44 | 3.02 |
| 17. | Flat tower | - - - | - - - | 0.88 | - - - |
| 18. | Globular echinate | 1.40 | 0.86 | 0.98 | 0.75 |
| 19. | Globular granulate | 9.39 | 6.95 | 2.70 | 5.03 |
| 20. | Globular polyhedral | 6.45 | 5.21 | 1.08 | 1.25 |
| 21. | Globular psilate | 2.38 | 3.91 | 0.09 | 1.00 |
| 22. | Horned tower | - - - | 1.30 | 0.68 | - - - |
| 23. | Oblong | - - - | - - - | 0.04 | - - - |
| 24. | Ovate | 4.06 | 3.47 | 1.08 | 0.50 |
| 25. | Parallelepipedal cell | 0.84 | 1.30 | 1.17 | 0.75 |
| 26. | Pentagon | - - - | 5.21 | - - - | - - - |
| 27. | Polyhedral | 3.64 | 0.43 | 0.63 | 1.51 |
| 28. | Polylobate | - - - | - - - | 1.32 | - - - |
| 29. | Prism | 0.28 | - - - | - - - | - - - |
| 30. | Rectangular | 1.68 | 4.34 | 0.93 | - - - |
| 31. | Reniform | - - - | - - - | 0.04 | - - - |
| 32. | Saddle | - - - | - - - | 56.48 | 2.77 |
| 33. | Saddle with notch | - - - | - - - | 0.73 | - - - |
| 34. | Scutiform | - - - | 0.43 | 0.14 | 0.75 |
| 35. | Sinuate elongate | - - - | - - - | 2.21 | 5.03 |
| 36. | Smooth elongate | 0.70 | 3.47 | 8.84 | 7.05 |
| 37. | Stellate | - - - | 0.86 | - - - | - - - |
| 38. | Tabular simple | 3.36 | 0.86 | 1.03 | - - - |
| 39. | Tabular irregular | 4.62 | 2.60 | - - - | - - - |
| 40. | Tabular polyhedral | 4.90 | 2.60 | 0.09 | 0.75 |
| 41. | Trapezoid | 3.08 | 4.78 | 2.31 | 10.57 |
| 42. | Triangular | 4.06 | 7.39 | 0.93 | 1.00 |
| 43. | Nodular | 0.70 | 2.60 | - - - | - - - |

Morphometric data on size dimensions (length, width, area, and perimeters) and shape descriptors (aspect ratio, solidity, and roundness) of morphotypes have been used for taxonomic resolution of plant species [48, 104, 117]. Morphometry of phytolith morphotypes that were present in high frequency was recorded (Tables 2–5). Data of size dimensions and shape

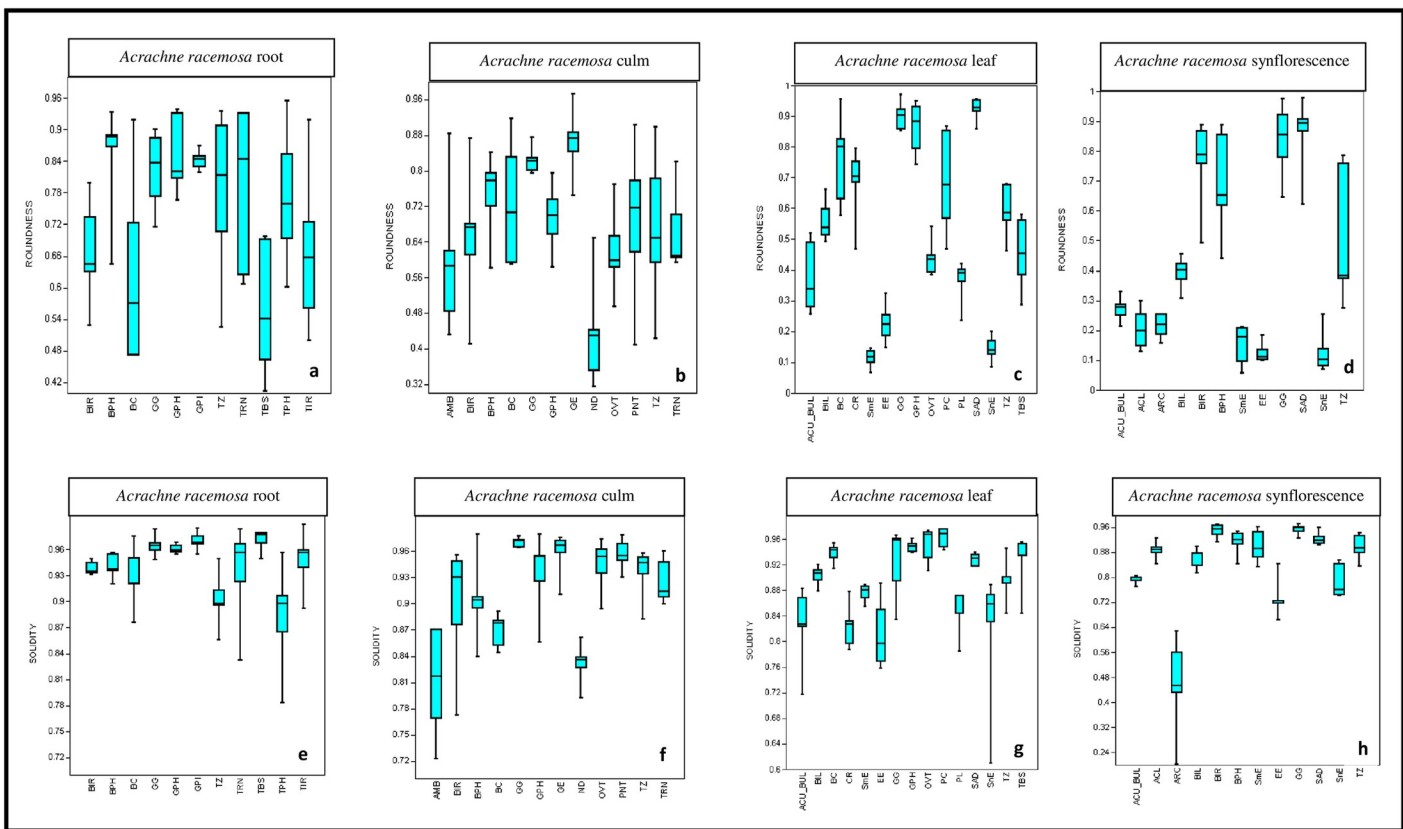

**Fig 8. Box-Whisker plots for median percentile and range of shape descriptors (roundness and solidity) of phytoliths in *Acrachne racemosa* (Heyne ex Roem. & Schult.) Ohwis.**

descriptors have been collected in our present study. Data of shape descriptors are shown in Fig 8. Solidity is a measure of the ratio between the total surface area to the convex area of a type and the roundness value is unity for perfect circles and with an increase in any of the dimensions, it decreases. When a particular morphotype is longer than wide, it shows that aspect ratio>unity and ranges from >1–3 for short cell types and >5 to <11 for long cell types. The roundness value of elongate echinate, smooth elongate, sinuate elongate does not exceed 0.30 (Fig 8c and 8d) and thus showed minimum roundness value. The roundness value of elongate types hardly exceeded 0.60 [71]. The globular polyhedral (Fig 8a and 8c) triangular, trapezoid (Fig 8a), ovate (Fig 8c), saddle (Fig 8c and 8d), and globular granular (Fig 8d) types of phytoliths showed maximum roundness values of >0.9. The globular psilate and tabular simple showed the maximum solidity value as seen in Box-Whisker plots (Fig 8e) whereas the minimum solidity value was shown by amoeboid and arculate (Fig 8h and 8f) type of phytoliths.

Correlation analysis among morphotypes of phytolith from different parts of *A. racemosa* based on Pearson's coefficient showed positive and negative correlation (Table 7). The strongest association was found between root and culm and the lowest between leaf and culm while root and synflorescence have a negative association. Leaf is positively associated with synflorescence. The leaf is more associated with synflorescence than root and culm. This shows that silica decomposition depends upon internal structures, physiology, and environmental factors (Table 7). Based on Jaccard's similarity index, the cluster diagram of phytolith morphotypes of species showed 67% of similarity between root and culm and were grouped whereas around

**Table 7. Coefficient correlation among the phytolith morphotypes from different parts of *Acrachne racemosa* (Heyne. Ex roem. & Schult.) Ohwibased on the Pearson's coefficient.**

| Species | | *Acrachne racemosa* | | | |
|---|---|---|---|---|---|
| | | **Root** | **Culm** | **Leaf** | **Synflorescence** |
| *Acrachne racemosa* | **Root** | 1 | 0.8092 | -0.0756 | -0.1103 |
| | **Culm** | 0.8092 | 1 | -0.0660 | -0.1238 |
| | **Leaf** | -0.0756 | -0.0660 | 1 | 0.1150 |
| | **Synflorescence** | -0.1103 | -0.1238 | 0.1150 | 1 |

58% of similarity was found between root, culm, and synflorescence. However, all parts of the *A. racemosa* showed 55% of similarities. The maximum similarity was found between root and culm (Fig 9).

## Biochemical architecture

**FTIR.** Through this technique, the purity of phytoliths has been examined [118–120]. FTIR spectra of silica from root, stem, leaves, and synflorescence of *A. racemosa* showed several peaks which were assigned to various structural units of silica. The wide peak is seen at 1088.94 cm$^{-1}$ in root and the longest peak is seen at 1058.93 cm$^{-1}$ in leaf. The peaks between 438.48–468.48 cm$^{-1}$ (Fig 10 and S1 Table) present in root, culm, leaf and synflorescence parts of the *A. racemosa* has been ascribed to deformation vibration of O-Si-O group [121] bonds. The peak at 665.94 cm$^{-1}$ has been assigned to symmetrical vibration of Si-O-Si [122] bonds present only in the leaves. The peaks between 786.94–801.46cm$^{-1}$ have been assigned to symmetrical vibration of Si-O [123] bonds are found in root, culm, leaf, and synflorescence. The peaks between 1058.93–1105.58 cm$^{-1}$ corresponds to the asymmetric vibration of Si-O-Si [124–126] bonds. The peak at 1875.88 cm$^{-1}$ has been assigned to deformation vibration of alkyl (R group) found only in the underground part (root). The peaks between 1633.69-1637-96 cm$^{-1}$ present only in root and culm could be ascribed to the deformation of H-O-H [127] bonds. The peaks between 2345.34–2362.47 cm$^{-1}$ and 3419.76–3425.90 cm$^{-1}$ correspond to the in-plane stretching vibration of Si-C [127] and O-H/Si-OH bonds [123] bonds.

## XRD analysis

Several polymorphic phases were seen depending upon the temperature and pressure [128]. Phytoliths isolated by XRD diffraction of different parts of *A. racemosa* showed peaks characteristics of different amorphous and crystalline polymorphic phases of silica that include Quartz, cristobalite alpha, tridymite, Quartz low, coesite, and stishovite (Fig 11). The shift from amorphous to crystalline phases of silica in cogon grass (*Imperata cylindrica* (L.) P. Beauv.) in the presence of potassium (K) was earlier reported by [129]. These phases have an identical chemical composition ($SiO_2$) but different lattice systems including anorthic, monoclinic, orthorhombic, hexagonal, and tetragonal. The present study shows that the tetragonal body-centered phase occurs only in culm, not in root, leaf and synflorescence which gives credence to the existence of polymorphic silica in plants [46, 103, 113, 115, 130]. XRD analysis of silica extracted from the grass species reported the presence of coesite, cristobalite and tridymite [131, 132]. Monoclinic primitive and hexagonal at non-ambient pressure phase were found only in synflorescence while orthorhombic body-centered and orthorhombic tridymite end-centered occurred in the root. Tridymite is derived from the oxidation of fayalite [133]. Peaks corresponding to tetragonal cristobalite alpha and anorthic primitive were found in all parts of the species while peak corresponding to monoclinic end-centered was found in culm

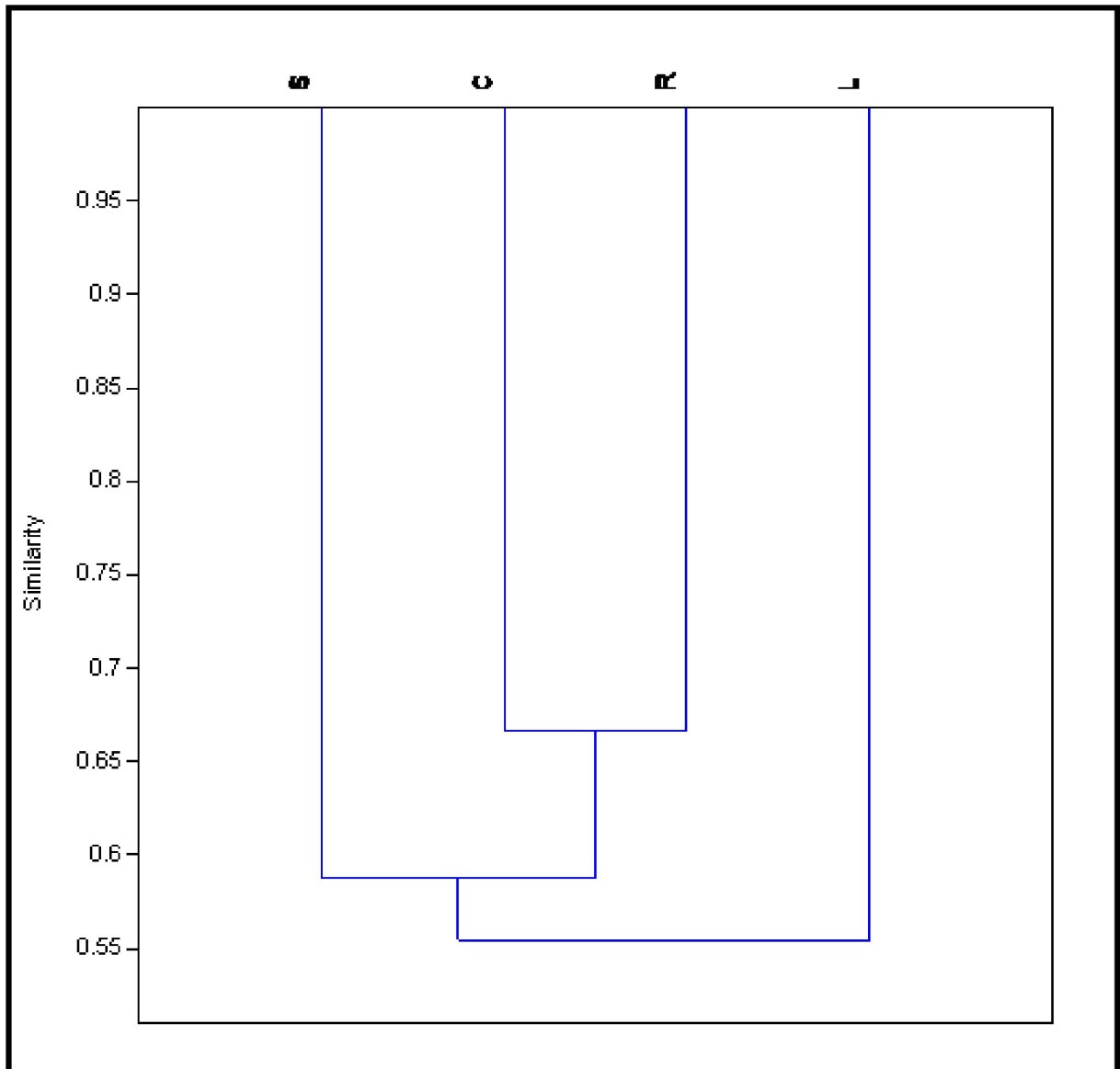

**Fig 9. Clustering of *Acrachne racemosa* (Heyne ex Roem. & Schult.) Ohwi based on presence/absence data of phytolith morphotypes.**
R = root, C = culm, L = leaf and S = synflorescence.

and root. Hexagonal quartz primitive were diagnostic of the root, culm and synflorescence. Hexagonal quartz low primitive, hexagonal primitive, monoclinic coesite end-centered and tetragonal stishovite primitive were present in root and synflorescence.

## Conclusion

*A. racemosa* shows a high frequency of saddle phytoliths that are transversely arranged in the costal area in the adaxial and abaxial region of the epidermal layer. The acute bulbous is present in the intercostal region in the abaxial surface while elongate echinate with concave ends are present in the intercostal region in the adaxial surface. Phytoliths isolated from aboveground and underground parts by the Dry Ashing Method show variety of phytolith

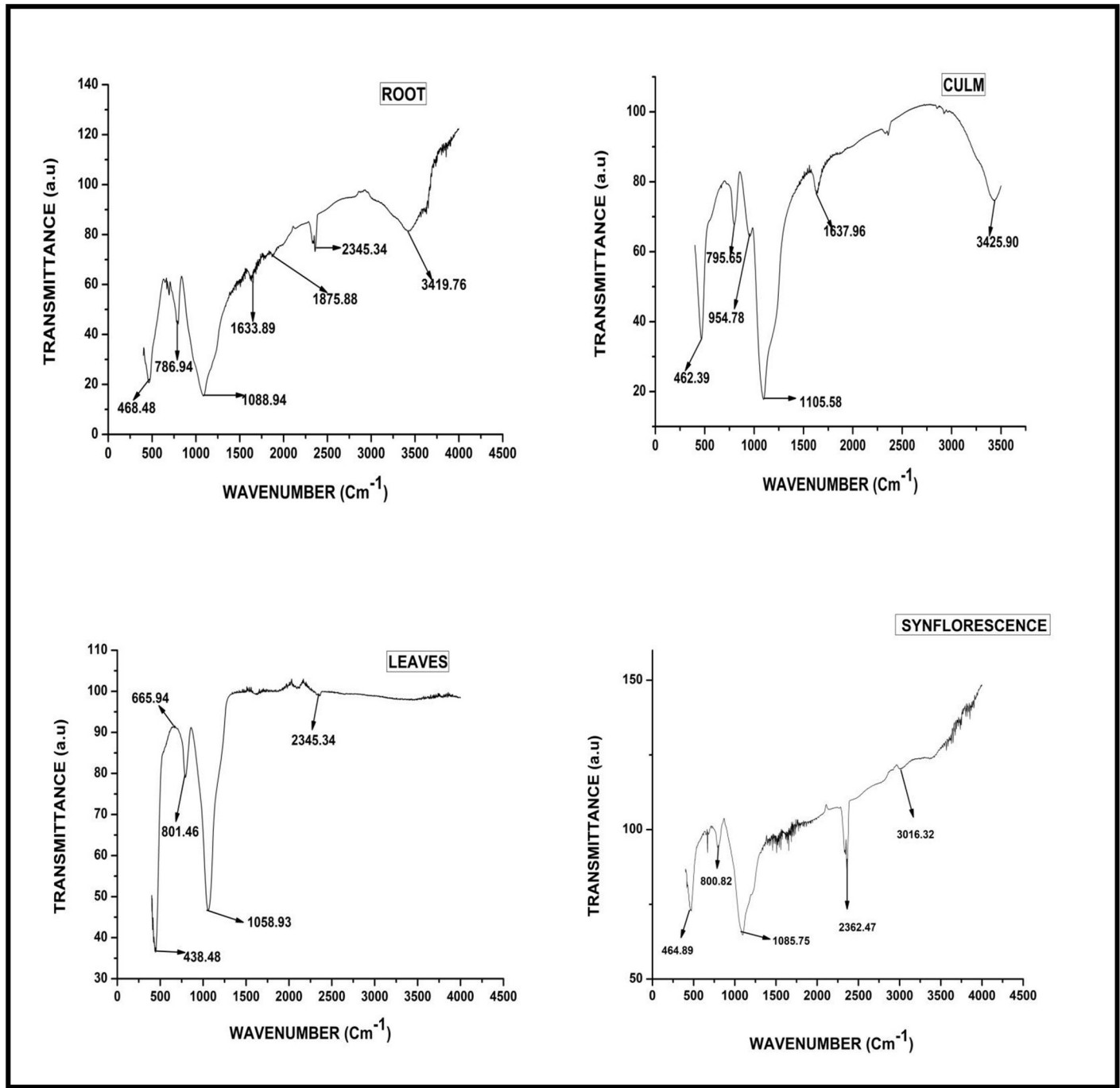

**Fig 10. FTIR spectra of phytoliths from different parts of *Acrachne racemosa* (Heyne ex Roem. & Schult.) Ohwi.**

morphotypes in terms of shape and size. Phytoliths present in roots are mostly blocky types. Various types of undulation patterns and ornamentations were seen in phytoliths from the fertile part i.e synflorescence. The vegetative part (leaf) possesses the highest amount of silica and ash followed by the underground part (root) than fertile part (synflorescence). The least amount of silica is possessed by the culm. Variation in silica amount, morphotypes, morphometric measurement and frequency depending on various factors involving anatomical,

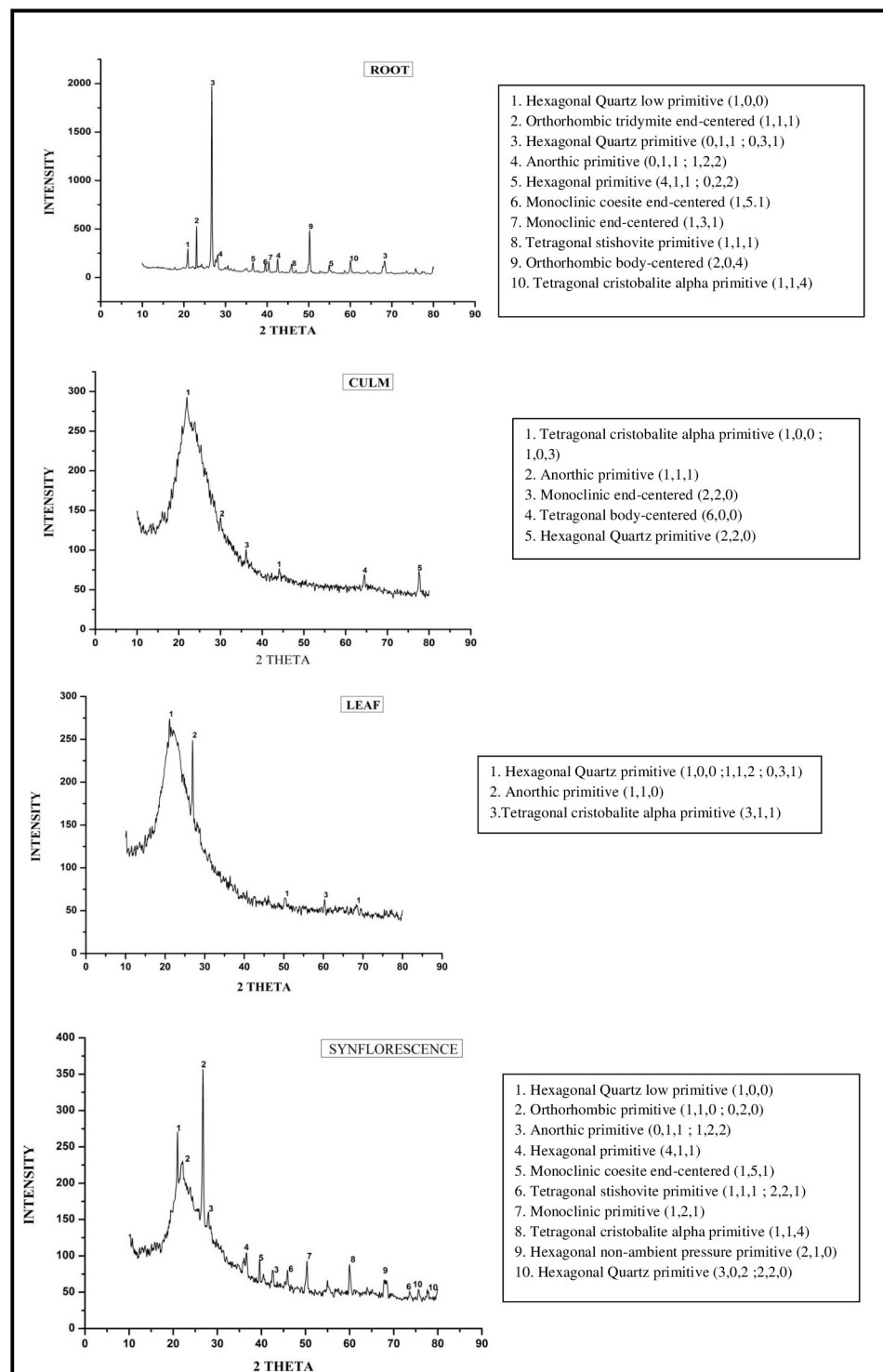

**Fig 11. XRD diffraction spectra of phytoliths isolated from different parts of *Acrachne racemosa* (Heyne ex Roem. & Schult.) Ohwi.**

physiological and environmental could be used for the identification and taxonomic demarcation of the species. Polylobate and cross-shaped types of phytoliths are characteristically found in leaf and Box-Whisker plots showed that Globular psilate and tabular simple possessed maximum solidity value. Blocky polyhedral possess the highest frequency in the root, blocky irregular in culm, saddle in leaf and undulations patterns complexed with different morphotypes have the highest frequency in synflorescence. The maximum similarity was found between root and culm. Phytoliths provide resistance and help the plant species in growth and development by protecting it from various stresses and play a very important role in research by acting as a tool for phytolith analysis in environmental and systematic biologists, geologists paleobotanistss and archeologists and help in the reconstruction and interpretation of the vegetation of grasslands. FTIR analysis showed various modes of vibrations. XRD diffraction of phytoliths showed peaks characteristics of different polymorphic phases of silica. The phytolith signatures of species developed in the present work shall provide additional parameters for the identification and diagnosis of the *A. racemosa*.

## Supporting information

**S1 Fig. Herbarium sheet of *Acrachne racemosa*(Heyne ex Roem. & Schult.) Ohwi.**
(TIF)

**S1 Table. FTIR peak showing different functional groups in phytoliths of *Acrachne racemosa*(Heyne ex Roem. & Schult.) Ohwi.**
(DOCX)

## Acknowledgments

The authors are thankful to the head of the Department of Chemistry, Guru Nanak Dev University, Amritsar for providing an XRD and FTIR instrumentation facility. We also thank Mr. Shakoor for his help in XRD analysis.

## Author Contributions

**Conceptualization:** Amarjit Singh Soodan.

**Data curation:** Priya Badgal.

**Formal analysis:** Priya Badgal, Poonam Chowdhary.

**Methodology:** Priya Badgal.

**Resources:** Priya Badgal, Mudassir Ahmad Bhat, Amarjit Singh Soodan.

**Supervision:** Amarjit Singh Soodan.

**Visualization:** Priya Badgal.

**Writing – original draft:** Priya Badgal.

**Writing – review & editing:** Priya Badgal, Amarjit Singh Soodan.

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
