## [Decision Letter · Decision Letter 0]

27 Apr 2021

PONE-D-21-09972

Phytolith profile of Acrachne racemosa (Heyne ex Roem. & Schult.) Ohwi (Cynodonteae, Chloridoideae, Poaceae)

PLOS ONE

Dear Dr. BADGAL,

Thank you for submitting your manuscript to PLOS ONE. After careful consideration, we feel that it has merit but does not fully meet PLOS ONE’s publication criteria as it currently stands. Therefore, we invite you to submit a revised version of the manuscript that addresses the points raised during the review process.

We look forward to receiving your revised manuscript.

Kind regards,

Umakanta Sarker

Academic Editor

PLOS ONE

Journal Requirements:

3. We note you have included a table to which you do not refer in the text of your manuscript. Please ensure that you refer to Table 3 in your text; if accepted, production will need this reference to link the reader to the Table.

Reviewers' comments:

Reviewer's Responses to Questions

**Comments to the Author**

1. Is the manuscript technically sound, and do the data support the conclusions?

Reviewer #1: Partly

Reviewer #2: Yes

2. Has the statistical analysis been performed appropriately and rigorously? 

Reviewer #1: Yes

Reviewer #2: Yes

3. Have the authors made all data underlying the findings in their manuscript fully available?

Reviewer #1: No

Reviewer #2: Yes

4. Is the manuscript presented in an intelligible fashion and written in standard English?

Reviewer #1: No

Reviewer #2: No

5. Review Comments to the Author

Reviewer #1: Manuscript Number: PONE-D-21-09972

Full Title: Phytolith profile of Acrachne racemosa (Heyne ex Roem. & Schult.) Ohwi (Cynodonteae, Chloridoideae, Poaceae)

The manuscript presents a very good, original & exhaustive piece of work done by the authors and the work has multidisciplinary applications. The phytoliths are very important taxonomic tools vis-à-vis aid in the reconstruction of paleo-environments and prediction of climate changes.

However, the manuscript needs a major revisionary work to make it worth accepting in PLOSONE Journal. Following are queries/points which need to be addressed and necessary changes should be incorporated in the manuscript accordingly.

1. The authority of the taxon under investigation is incomplete.

2. The abstract of the manuscript is not coherent & robust and needs to be re-written.

3. The manuscript entitled “A worldwide phylogenetic classification of the Poaceae (Gramineae) II: An update and a comparison of two 2015 classifications.” which has been cited in the abstract section as [2] has been wrongly interpreted as the number of sub-tribes under the tribe cyanodontae are 21 not 18.

4. The introduction needs a systematic arrangement of sentences as it lacks logical sequence of sentences.

5. The manuscript should be supplemented by the herbarium sheet of the taxon under investigation.

6. The methodology should be made brief and concise as the manuscript is already very much voluminous and lengthy.

7. The methodology for FTIR is not clear.

8. The methodology for XRD is not mentioned instead use of XRD is presented.

9. Silica is an amorphous material but you have observed the crystalline phases in XRD studies. Explain the reasons for this shift from amorphous to crystalline behavior of silica.

10. There is repetition of sentences like---- Acrachne racemosa (Heyne. Ex roem. & Schult.) Ohwi is an annual grass having erect, simple or branched, tufted culms and…..the repetitions of sentences should be avoided.

11. There is no need of the Table 1 an this information already exists in literature vis-à-vis on various taxonomic compilations and e-floras.

12. Explain the reasons for the accumulation of more silica content in leaves (The silica is absorbed by the plant roots from the soil and is then carried along the transpirational stream to different plant parts where it is subsequently deposited----correlate the silica content of various plant parts in relation to this statement)

13. The presence of saddle shaped phytoliths is characteristic feature of subfamily cyanodonteae……no such mention has been made in the manuscript despite of the presence of saddles in leaf peelings.

14. The results of In-situ locations of phytoliths should be rewritten so as to make then concise, brief and meaningful.

15. In Fig.2…Adaxial (B) should be presented before the Abaxial (A).

16. Incorporate the Fig. numbers in the morphometry tables of phytoliths against the each phytolith morphotype presented in morphometric tables.

17. The morphotypes in tables with and without data does not correlate with the data presented in the text. There is lot of ambiguity in the morphotype data (In Table 2A 27 morphotypes are mentioned from root.....data of only 11 morphotypes is presented and in text only 24 morphotypes in mentioned in total from the root).

18. ) There is also a lot of ambiguity in the figure number of morphtypes between the text and the legend of the figures.

19. No methodology for the study of undulations of phytoliths is presented. Mention the respective parts of the synflorescence from which these undulation patterns were reported (glumes, lemma, palea etc)…if not worked out in the present study…….give support from literature

20. The cumulative frequency data of all the phytolith morphotypes from a plant part should be equal to 100 or 100% but the values are lower that 100% which is inappropriate.

21. Present the FTIR RESULTS in a tabulated form.

22. The transition of amorphous silica to crystalline form does not take place at 550 0C but at a temperature higher than 10000C. Support your finding from literature and look for some other reasons that may have brought about phase shift from amorphous to crystalline form of silica.

23. NO mention is made in the manuscript about the method of deciphering of XRD peaks. Have you made use of literature in peak deciphering or any software package and if so send us the working pics of the methods of peak deciphering for surety of the results presented in the manuscript.

Thanks and best of luck

Reviewer #2: Dear Authors

I ask you to proofread the text for grammatical errors.

The main question on the article - what in your opinion is the biological role of shapeless phytoliths (Fig. A, B, D)? May be you will find some ideas in:

https://www.sciencedirect.com/science/article/abs/pii/S036725301830495X?via%3Dihub

https://www.hindawi.com/journals/bmri/2014/648326/

https://besjournals.onlinelibrary.wiley.com/doi/full/10.1111/1365-2435.12692

https://www.mdpi.com/2223-7747/8/8/249

6. PLOS authors have the option to publish the peer review history of their article (what does this mean?). If published, this will include your full peer review and any attached files.

Reviewer #1: **Yes: **SHEIKH ABDUL SHAKOOR, PG DEPARTMENT OF BOTANY, THE ISLAMIA COLLEGE OF SCIENCE AND COMMERCE, SRINAGAR (J&K) INDIA

Reviewer #2: No

---

## [Author Response · Author response to Decision Letter 0]

20 Oct 2021

The Editor-in-Chief

PLOS ONE

Dear Sir

I have uploaded the files for Response to Reviewers in the attachments.

---

## [Decision Letter · Decision Letter 1]

10 Dec 2021

PONE-D-21-09972R1

Phytolith profile of Acrachne racemosa (B. Heyne ex Roem. & Schult.) Ohwi (Cynodonteae, Chloridoideae, Poaceae)

PLOS ONE

Dear Dr. BADGAL,

Thank you for submitting your manuscript to PLOS ONE. After careful consideration, we feel that it has merit but does not fully meet PLOS ONE’s publication criteria as it currently stands. Therefore, we invite you to submit a revised version of the manuscript that addresses the points raised during the review process.

The authors addressed all the comments raised by reviewers and the one reviewer is accepted the MS and another reviewer declined invitation due to his bussiness. Now, the manuscript improved substantially. However, before its acceptance, the authors should address these issues (Typos errors) again with minor revision.

Line 68: change “ha-1” to “ha^-1^”. [superscript (-1)]. Follow this style throughout the whole MS where it exists

Line 122: change “50ml” to “50 mL”. change “80◦C” to degree symbol “80 °C”. Follow this style throughout the whole MS where it exists

Line 145: change letter “x” to symbol of cross “×”

Line 221: change “bar =40” to “bar = 40”. Add spaces before and after the symbol “=”. Follow this style throughout the whole MS where it exists

Table 2a, 2b, 2c, 2d: Add spaces before and after the symbol “±”.

Line 367: delete the space after the slash “/”

Line 443: add spaces before and after the symbol “>” and “˂”

We look forward to receiving your revised manuscript.

Kind regards,

Umakanta Sarker

Academic Editor

PLOS ONE

Journal Requirements:

Reviewers' comments:

Reviewer's Responses to Questions

**Comments to the Author**

1. If the authors have adequately addressed your comments raised in a previous round of review and you feel that this manuscript is now acceptable for publication, you may indicate that here to bypass the “Comments to the Author” section, enter your conflict of interest statement in the “Confidential to Editor” section, and submit your "Accept" recommendation.

Reviewer #2: All comments have been addressed

2. Is the manuscript technically sound, and do the data support the conclusions?

Reviewer #2: Yes

3. Has the statistical analysis been performed appropriately and rigorously? 

Reviewer #2: Yes

4. Have the authors made all data underlying the findings in their manuscript fully available?

Reviewer #2: Yes

5. Is the manuscript presented in an intelligible fashion and written in standard English?

Reviewer #2: Yes

6. Review Comments to the Author

Reviewer #2: Your paper is devoted to very interesting and "hot" topic - biomineralization in plants.

Nowdays we need more data-based phytolith papers.

Please, go forward!

7. PLOS authors have the option to publish the peer review history of their article (what does this mean?). If published, this will include your full peer review and any attached files.

Reviewer #2: No

---

## [Author Response · Author response to Decision Letter 1]

24 Jan 2022

I have attached the file as "Response to reviewers".

Line 68: change “ha-1” to “ha-1”. [superscript (-1)]. Follow this style throughout the whole MS wherever it exists

Ans: “ha-1” has been corrected to “ha-1” (Line 69) and this style has been followed throughout the whole MS wherever it exists in the revised manuscript.

Line 122: change “50ml” to “50 mL”. change “80◦C” to degree symbol “80 °C”. Follow this style throughout the whole MS where it exists

Ans: “50ml” has been corrected to “50 mL” and 80◦C to 80 °C (Line 123) and this style has been followed throughout the whole MS wherever it exists in the revised manuscript.

Line 145: change letter “x” to symbol of cross “×”

Ans: Letter “x” has been changed to symbol of cross “×” in the revised manuscript (Line 146).

Line 221: change “bar =40” to “bar = 40”. Add spaces before and after the symbol “=”. Follow this style throughout the whole MS where it exists

Ans: “bar =40” has been changed to “bar = 40” (Line 222) and the spaces has been added before and after the symbol “=”. This style has been followed throughout the whole MS wherever it exists in the revised manuscript.

Table 2a, 2b, 2c, 2d: Add spaces before and after the symbol “±”.

Ans: Spaces have been added before and after the symbol “±” in table 2a, 2b, 2c, 2d in the revised manuscript.

Line 367: delete the space after the slash “/”

Ans: Space has been deleted in the revised manuscript (Line 363).

Line 443: add spaces before and after the symbol “>” and “˂”

Ans: Spaces have been added before and after the symbol “>” and “˂” in the revised manuscript (Line 410).

---

## [Editor Report · Decision Letter 2]

26 Jan 2022

Phytolith profile of Acrachne racemosa (B. Heyne ex Roem. & Schult.) Ohwi (Cynodonteae, Chloridoideae, Poaceae)

PONE-D-21-09972R2

Dear Dr. BADGAL,

We’re pleased to inform you that your manuscript has been judged scientifically suitable for publication and will be formally accepted for publication once it meets all outstanding technical requirements.

Kind regards,

Umakanta Sarker

Academic Editor

PLOS ONE

Additional Editor Comments (optional):

During proofreading: address the following errors:

Line 60: Change “SiO2.nH2O” to “SiO2.nH2O”. (Subscript the number)

Line 111: Change “31.31 ºN and 74.55 ºE.” to “31.31ºN and 74.55ºE.”.

Line 150: Change “0.1mg” to “0.1 mg”.

Table 2a, 2b, 2c, 2d: reduce font size to accommodate all values in a single row.

References: Follow the journal style
---

## [Editor Report · Acceptance letter]

2 Feb 2022

PONE-D-21-09972R2 

Phytolith profile of *Acrachne racemosa* (B. Heyne ex Roem. & Schult.) Ohwi (Cynodonteae, Chloridoideae, Poaceae) 

Dear Dr. Badgal:

I'm pleased to inform you that your manuscript has been deemed suitable for publication in PLOS ONE. Congratulations! Your manuscript is now with our production department. 

Kind regards, 

on behalf of

Professor Umakanta Sarker 

Academic Editor

PLOS ONE